# Higher skeletal muscle mitochondrial oxidative capacity is associated with preserved brain structure up to over a decade

Qu Tian [1] ✉, Erin E. Greig [1], Christos Davatzikos [2], Bennett A. Landman [3,4], Susan M. Resnick [5] & Luigi Ferrucci [1]

Impaired muscle mitochondrial oxidative capacity is associated with future cognitive impairment, and higher levels of PET and blood biomarkers of Alzheimer's disease and neurodegeneration. Here, we examine its associations with up to over a decade-long changes in brain atrophy and microstructure. Higher in vivo skeletal muscle oxidative capacity via MR spectroscopy (post-exercise recovery rate, $k_{PCr}$) is associated with less ventricular enlargement and brain aging progression, and less atrophy in specific regions, notably primary sensorimotor cortex, temporal white and gray matter, thalamus, occipital areas, cingulate cortex, and cerebellum white matter. Higher $k_{PCr}$ is also associated with less microstructural integrity decline in white matter around cingulate, including superior longitudinal fasciculus, corpus callosum, and cingulum. Higher in vivo muscle oxidative capacity is associated with preserved brain structure up to over a decade, particularly in areas important for cognition, motor function, and sensorimotor integration.

Aging has profound effects on human brains at the molecular, cellular, morphological, and functional levels. Due to their postmitotic nature, neurons depend on mitochondrial quality control to maintain bioenergetic demands[1,2]. Dysfunctional mitochondria may cause neuronal dysfunction and death through different mechanisms, including reduced adenosine triphosphate (ATP) production, supraphysiological levels of reactive oxygen species (ROS), defects in calcium handling, and defects in other less studied physiological mechanisms[3,4]. Neurons also rely on support from non-neuronal glial cells for their energy requirements, mainly including astrocytes, microglia, and oligodendrocytes, where mitochondria are thought to play a key role[5]. Mitochondria in astrocytes are responsible for oxidative metabolism, acting as storage organelles for calcium ions ($Ca^{2+}$), and key to intercellular signaling. Healthy mitochondria govern microglial activation in response to neuroinflammation as the first line of defense in the central nervous system (CNS). Myelination of oligodendrocytes requires enormous amounts of energy supported by healthy mitochondria to ensure efficient neuronal communication[6].

The mitochondrial cascade hypothesis, first proposed in 2004, centers on the key role of age-related mitochondrial decline in Alzheimer's disease (AD), especially late-onset AD[7]. Although animal and human cell culture studies provide evidence supporting this hypothesis, in vivo human data from the human population level has just started to emerge[8]. A major obstacle in this research field is the difficulty of studying mitochondrial oxidative capacity in the in vivo human brain. A possible alternative is to consider skeletal muscle mitochondria oxidative capacity as a proxy measure of overall mitochondria oxidative capacity. Our recent findings show that in vivo

[1]Longitudinal Studies Section, Translational Gerontology Branch, National Institute on Aging, Baltimore, MD, USA. [2]Radiology Department, Perelman School of Medicine, University of Pennvalnia, Philadelphia, PA, USA. [3]Department of Computer Science, Vanderbilt University, Nashville, TN, USA. [4]Department of Electrical and Computer Engineering, Vanderbilt University, Nashville, TN, USA. [5]Brain Aging and Behavior Section, Laboratory of Behavioral Neuroscience, National Institute on Aging, Baltimore, MD, USA. ✉e-mail: qu.tian@nih.gov

**Table 1 | Participant characteristics at the index visit (*n* = 649)**

| | Overall sample (n = 649) | With longitudinal data (n = 463) | With cross-sectional data only (n = 186) |
|---|---|---|---|
| | Mean ± standard deviation, [range], or N (%) as noted | | |
| **Demographics** | | | |
| Age, years | 65.9 ± 15.4 [22.4–97.7] | 70.9 ± 12.3 [30.9–97.7] | 53.5 ± 15.2 [22.4–90.7] |
| Women | 365 (56%) | 259 (56%) | 106 (57%) |
| Black | 161 (25%) | 125 (27%) | 36 (19%) |
| Apolipoprotein E ε4 carriers | 141 (25%) (n = 553) | 113 (25%) (n = 446) | 28 (26%) (n = 107) |
| **Physical activity** | | | |
| Inactive | 46 (7%) | 45 (10%) | 11 (6%) |
| Low | 277 (43%) | 202 (44%) | 75 (40%) |
| Moderate | 194 (30%) | 140 (30%) | 54 (29%) |
| Very active | 122 (20%) | 76 (16%) | 46 (25%) |
| **Fitness** | | | |
| 400-meter walk time, s | 262.2 ± 49.8 [156.3–612.8] (n = 627) | 269.8 ± 52.7 [156.3–612.8] (n = 442) | 244.1 ± 36.3 [156.9–397.5] (n = 185) |
| VO$_2$ max, mL/kg/min | 23.9 ± 7.7 [3.6–54.8] (n = 586) | 22.4 ± 6.7 [3.6–52.1] (n = 417) | 27.7 ± 8.6 [6.0–54.8] (n = 169) |
| **Skeletal muscle oxidative capacity (k$_{PCr}$, s$^{-1}$)** | 0.0218 ± 0.0057 [0.0101–0.0541] | 0.0213 ± 0.0055 [0.0101–0.0541] | 0.0231 ± 0.0059 [0.0124–0.0449] |
| Total number of brain MRI | 1842 | 1656 | 186 |
| 1 | 186 (29%) | - | - |
| 2 | 145 (22%) | - | - |
| 3 | 110 (17%) | - | - |
| 4 | 98 (15%) | - | - |
| 5 | 58 (9%) | - | - |
| 6 | 27 (4%) | - | - |
| 7 | 17 (2%) | - | - |
| 8–11 | 8 (1%) | - | - |
| Average follow-up, years | 4.5 ± 2.3 [1–11] | 4.5 ± 2.3 [1–11] | - |
| Average number of visits per participant | 3 ± 1.8 [1–11] | 3.6 ± 1.6 [1–11] | 1 ± 0 |

Note: Of 649 participants with brain magnetic resonance imaging (MRI) data, 639 had diffusion tensor imaging (DTI) data with a total of 1,718 DTI assessments. The first concurrent assessment visit of skeletal muscle oxidative capacity and brain MRI scans was determined as the index visit.

skeletal muscle oxidative capacity via MR spectroscopy predicts the development of cognitive impairment or dementia and is associated with brain PET and blood biomarkers of AD and neurodegeneration[9]. Mitochondrial DNA copy number from blood samples predicts future dementia and is associated with cognitive function[10–12]. The study of longitudinal changes in brain structure associated with mitochondrial health may deepen our insights into mechanisms that connect mitochondria and brain health. However, data on how in vivo assessment of mitochondrial function is associated with brain structure are sparse. One recent study that examined the relationship of blood mitochondria DNA copy number with brain structure failed to demonstrate a significant relationship, perhaps because of cross-sectional design, assessments limited to gross morphometry of the brain structure (such as total brain volume or white matter hyperintensities) and a lack of in vivo assessment of mitochondria[12]. We previously reported cross-sectional associations of skeletal muscle oxidative capacity with PET imaging markers of amyloid in the brain[9] and longitudinal brain atrophy with dual decline in memory and gait related to mitochondrial

dysfunction[13,14]. The associations were found in several brain regions but strong in specific frontal and parietal areas, cingulate cortex, and cerebellum[9,13].

In this study, we aimed to expand prior research by examining the relationships between in vivo skeletal muscle mitochondrial function and up to over a decade-long changes in brain structure, including volumes and microstructural integrity of gray matter and white matter. By leveraging longitudinal neuroimaging data collected up to over a decade from community-dwelling adults, we aimed to answer the following questions: (1) is higher muscle mitochondrial function associated with preserved brain structure over time during aging? (2) are there areas of the brain that are particularly affected by muscle mitochondrial oxidative capacity? (3) is the gray matter or white matter more strongly associated with muscle mitochondrial oxidative capacity?

## Results
Participant characteristics are presented in Table 1. Selection of the study sample is shown in Fig. 1. The average age of these 649 participants was 65.9 years at the index visit and 70% were aged 60 years or older. 56% were women, 25% were Black, and 91% were right-handed. 463 of the 649 participants had repeated brain MRI scans with the follow-up interval ranging from 1 to 11 years.

### Skeletal muscle oxidative capacity and brain volumes via MRI
There were few cross-sectional associations between k$_{PCr}$ and brain volumes (Table 2, Fig. 2). k$_{PCr}$ was only cross-sectionally associated with the overall ventricular volume, the lateral and 4$^{th}$ ventricles, and volumes of the medial frontal cortex, occipital fusiform gyrus, and occipital white matter ($p = 0.025$, 0.029, 0.037, 0.039, 0.006, and 0.018, respectively) (Table 2, Fig. 2). None of these survived corrections for multiple testing (all FDR-adjusted $p > 0.05$).

Longitudinally, higher k$_{PCr}$ was marginally associated with slower enlargement of the total ventricular volume and was significantly associated with slower worsening in 3$^{rd}$ and 4$^{th}$ ventricular volumes and SPARE-BA (brain aging) scores ($p = 0.073$, 0.041, 0.010, and 0.011, respectively) (Table 2, Fig. 2). Longitudinal associations with worsening in 4$^{th}$ ventricular volume and SPARE-BA scores also survived corrections for multiple testing using FDR ($\beta = -0.003$, FDR-adjusted $p = 0.042$; and $\beta = -0.005$, FDR-adjusted $p = 0.042$, respectively). Higher k$_{PCr}$ was significantly associated with slower rates of gray matter atrophy in several regions, including frontal, parietal, temporal, occipital, cingulate, and basal ganglia-related areas (Table 2, Fig. 2). Notably, most of these longitudinal associations remained statistically significant after correcting for multiple testing using FDR, including the inferior frontal gyrus, posterior orbital gyrus, subcallosal area, operculum, precentral gyrus, entorhinal cortex, parahippocampal gyrus, middle temporal gyrus, superior, medial, and middle occipital gyri, occipital fusiform gyrus, and anterior and middle cingulate cortex (all FDR-adjusted $p < 0.042$; Table 2, Fig. 2). Results remained similar after adjusting for total brain volume (Supplementary Fig. 1), and after removing data points at and after cognitive impairment (Supplementary Fig. 2). Longitudinal associations with brain volume change remained similar after adjusting for 400-meter walk time, VO$_2$ max, and apolipoprotein E ε4 carrier status (Supplementary Table 1). Results remained largely unchanged when the analysis was limited to participants aged 60 years or older (Supplementary Fig. 3).

### Skeletal muscle oxidative capacity and gray matter microstructure via DTI
There were no cross-sectional associations with DTI measures in gray matter except mean diffusivity (MD) in the occipital pole (Supplementary Fig. 4). There were no longitudinal associations between skeletal muscle mitochondrial oxidative capacity and gray matter

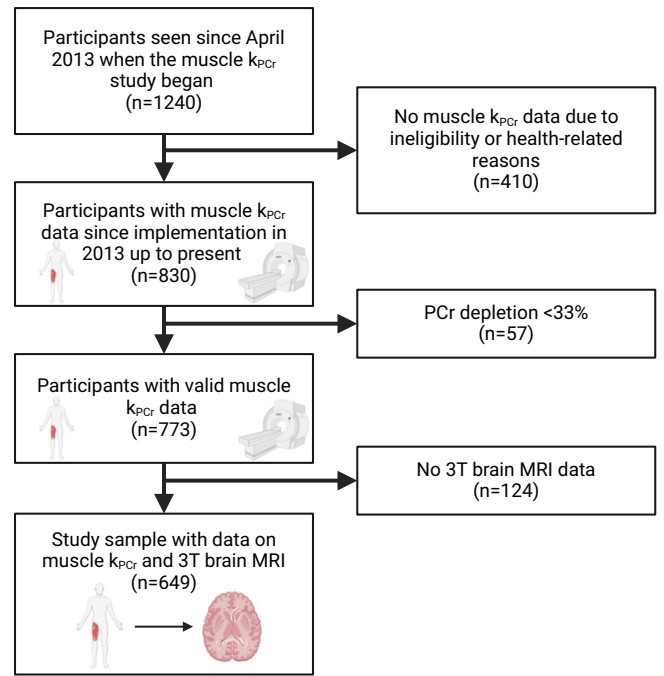

**Fig. 1 | Flow chart of the selection of study sample.** Legend: MRI magnetic resonance imaging. Created in BioRender. Greig, E. (2024) https://BioRender.com/k39x234. $k_{PCr}$ stands for the post-exercise recovery rate of phosphocreatine (PCr).

microstructure using fractional anisotropy (FA) or MD (Supplementary Fig. 4) (all $p > 0.05$).

### Skeletal muscle oxidative capacity and white matter microstructure via DTI

Cross-sectionally, higher $k_{PCr}$ was associated with higher FA and lower MD in the total white matter ($p = 0.029$ and $p = 0.023$, respectively). Higher $k_{PCr}$ was also cross-sectionally associated with higher FA and lower MD and RD in the splenium of the corpus callosum, and lower MD and AD in the hippocampal part of the cingulum (Table 3, Fig. 3, Supplementary Table 2).

Longitudinally, $k_{PCr}$ was not associated with the change in FA or MD of the total white matter ($p = 0.064$ and $p = 0.141$, respectively). Higher $k_{PCr}$ was associated with slower rates of decline of FA in the body of the corpus callosum and superior longitudinal fasciculus ($\beta = 0.007$, $p = 0.046$ and $\beta = 0.008$, $p = 0.044$, respectively) and these associations did not survive corrections for multiple testing using FDR (Table 3, Fig. 3). $k_{PCr}$ was marginally associated with slower declines of FA in the splenium of the corpus callosum, the cingulate and hippocampal part of the cingulum and fornix (cres)/ stria terminalis ($p = 0.065$, $0.064$, $0.065$, and $0.058$, respectively) (Table 3, Fig. 3). Higher $k_{PCr}$ was also associated with less increase in RD of the hippocampal part of the cingulum ($\beta = -0.011$, $p = 0.036$) (Supplementary Table 2). Results remained similar after removing data points at and after cognitive impairment (Supplementary Fig. 5). Longitudinal associations with changes in FA and MD of white matter remained similar after adjusting for 400-meter walk time and APOE ε4 carrier status but were attenuated after adjusting for $VO_2$ max (Supplementary Table 3). Results remained largely unchanged among participants aged 60 years or older (Supplementary Fig. 6).

Brain areas that showed significant longitudinal associations with $k_{PCr}$ are shown in Fig. 4. Estimated trajectories of top significant MRI and DTI markers among participants with higher and lower levels of $k_{PCr}$ as well as a random sample of individual datapoints are shown in Supplementary Fig. 7. In the exploratory analysis of sex differences, for most neuroimaging measures of MRI and DTI, there was no significant interaction term with sex in longitudinal associations, except MD of the cingulate part of the cingulum and superior corona radiata neither of which passed FDR-adjusted $p < 0.05$. For these 2 MD measures, we further stratified by sex. The longitudinal associations with MD of the cingulate part of the cingulum and superior corona radiata showed a trend in men ($p = 0.052$ and $0.099$, respectively), but were not significant in women ($p = 0.952$ and $0.301$, respectively).

## Discussion

This longitudinal study demonstrates a significant relationship between skeletal muscle mitochondrial oxidative capacity and brain structural changes up to over a decade, emphasizing the strong connection between mitochondrial health and brain aging and neurodegeneration. By investigating two different neuroimaging modalities across multiple brain regions, we identified specific brain regions and connecting tracts that were related to mitochondrial oxidative capacity assessed in the skeletal muscle. These longitudinal findings provide mechanistic insights into the connection between muscle bioenergetics and brain aging and lay a foundation for future research on mitochondrial bioenergetics in the brain.

Skeletal muscle mitochondrial function showed strong associations with longitudinal brain changes, including global and regional brain atrophy, spanning up to 11 years of neuroimaging data. In contrast to our longitudinal findings, the lack of cross-sectional associations may be due to greater inter-individual variability at one-time assessment. Also, there may be a discrepancy in time between biological dysfunction (e.g., mitochondria) and phenotypic (e.g., brain structure) and functional (e.g., dementia) impairments because of functional reserve and resilience mechanisms[15]. The lack of cross-sectional associations may be hampered by this problem. The findings of this study are consistent with our previous work connecting mobility and neuroimaging outcomes where we did not detect cross-sectional associations but found strong, significant longitudinal associations[13,16]. Because the majority of the study participants are aged 60 years or older, whether this relationship persists in younger age requires future research.

The investigation of multiple cortical and subcortical areas across the whole brain reveals locations in the brain that appear to be most sensitive to differences in muscle mitochondrial oxidative capacity. Among significant longitudinal associations, top-ranked brain areas included the primary sensorimotor cortex, occipital lobe, precuneus, cingulate cortex, insula, temporal areas, and white matter tracts, particularly the association fibers and commissural fibers of the corpus callosum. Notably, amyloid tracer uptake in the primary sensorimotor cortex, cingulate cortex, and other temporal areas is also strongly associated with muscle mitochondrial function[9]. Taken together, these data suggest that muscle mitochondrial function connects to both neuropathology and morphological changes in specific brain areas. Interestingly, the gray matter regions identified in this study are important for sensorimotor integration, cognition, motor function, and coordination. Thus, these results are consistent with recent findings on the association between sarcopenia and brain structure[17,18]. Some of these regions appear to fall along the cortical and internal border zones where infarcts often occur due to hypoperfusion or microemboli[19,20]. Whether the connection between mitochondrial dysfunction and brain atrophy is through impaired perfusion and microcirculation warrants further investigation.

Longitudinal changes in specific white matter tracts, including the corpus callosum and association fibers of the superior longitudinal fasciculus and cingulum, showed associations with muscle mitochondrial function. The association fibers, including the superior longitudinal fasciculus and cingulum, anatomically and functionally link gray matter areas in the same hemisphere and play key roles in memory, attention, and other cognitive functions[21,22], whereas the corpus callosum connects gray matter areas from the opposite

**Table 2 | Associations between skeletal muscle oxidative capacity and brain volumes via MRI (n = 649)**

| | | Cross-sectional associations | | | | Longitudinal associations | | | |
|---|---|---|---|---|---|---|---|---|---|
| | | β | SE | *p*-value | FDR-adj. *p* | β | SE | *p*-value | FDR-adj. *p* |
| **Global** | SPARE-BA (brain aging) scores | −0.0427 | 0.0226 | 0.0596 | 0.5189 | −0.0052 | 0.0020 | **0.0109** | **0.0417** |
| | SPARE-AD (Alzheimer's disease) scores | −0.0209 | 0.0372 | 0.5743 | 0.8759 | −0.0053 | 0.0035 | 0.1302 | 0.1891 |
| **Ventricles** | Ventricle | −1.5795 | 0.7026 | **0.0249** | 0.3987 | −0.0678 | 0.0378 | 0.0731 | 0.1350 |
| | Lateral ventricle | −1.4327 | 0.6558 | **0.0293** | 0.3987 | −0.0581 | 0.0346 | 0.0932 | 0.1497 |
| | 3rd ventricle | −0.0346 | 0.0221 | 0.1175 | 0.6791 | −0.0024 | 0.0012 | **0.0414** | 0.0942 |
| | 4th ventricle | −0.0516 | 0.0247 | **0.0374** | 0.3987 | −0.0026 | 0.0010 | **0.0097** | **0.0417** |
| **Frontal** | Frontal pole | 0.0175 | 0.0311 | 0.5736 | 0.8759 | 0.0023 | 0.0023 | 0.3054 | 0.3726 |
| | Superior frontal gyrus | −0.0693 | 0.0928 | 0.4557 | 0.8759 | 0.0109 | 0.0073 | 0.1350 | 0.1908 |
| | Middle frontal gyrus | −0.0561 | 0.1162 | 0.6296 | 0.8852 | 0.0130 | 0.0084 | 0.1223 | 0.1819 |
| | Orbital part of inferior frontal gyrus | 0.0122 | 0.0202 | 0.5450 | 0.8759 | 0.0013 | 0.0012 | 0.2933 | 0.3678 |
| | Triangular part of inferior frontal gyrus | 0.0601 | 0.0389 | 0.1235 | 0.6791 | 0.0015 | 0.0020 | 0.4701 | 0.5030 |
| | Opercular part of inferior frontal gyrus | −0.0253 | 0.0352 | 0.4726 | 0.8759 | 0.0049 | 0.0019 | **0.0103** | **0.0417** |
| | Medial frontal cortex | 0.0427 | 0.0207 | **0.0392** | 0.3987 | 0.0021 | 0.0012 | 0.0875 | 0.1443 |
| | Anterior orbital gyrus | 0.0215 | 0.0182 | 0.2361 | 0.8472 | 0.0011 | 0.0013 | 0.3987 | 0.4504 |
| | Posterior orbital gyrus | 0.0257 | 0.0278 | 0.3562 | 0.8759 | 0.0045 | 0.0015 | **0.0033** | **0.0416** |
| | Lateral orbital gyrus | 0.0250 | 0.0210 | 0.2331 | 0.8472 | 0.0013 | 0.0014 | 0.3462 | 0.4141 |
| | Medial orbital gyrus | 0.0065 | 0.0284 | 0.8182 | 0.9266 | 0.0042 | 0.0022 | 0.0548 | 0.1113 |
| | Supplementary motor area | 0.0331 | 0.0444 | 0.4554 | 0.8759 | 0.0029 | 0.0037 | 0.4278 | 0.4660 |
| | Subcallosal area | −0.0253 | 0.0155 | 0.1024 | 0.6791 | 0.0027 | 0.0010 | **0.0089** | **0.0416** |
| | Insula | 0.0493 | 0.0464 | 0.2888 | 0.8759 | 0.0079 | 0.0035 | **0.0224** | 0.0718 |
| | Operculum | −0.0015 | 0.0553 | 0.9789 | 0.9844 | 0.0103 | 0.0039 | **0.0088** | **0.0416** |
| | Precentral gyrus | 0.0607 | 0.0870 | 0.4852 | 0.8759 | 0.0166 | 0.0057 | **0.0035** | **0.0416** |
| | Frontal white matter | 0.2403 | 0.4641 | 0.6048 | 0.8852 | 0.0520 | 0.0319 | 0.1036 | 0.1620 |
| **Parietal** | Postcentral gyrus | 0.0448 | 0.0744 | 0.5469 | 0.8759 | 0.0105 | 0.0045 | **0.0215** | 0.0718 |
| | Superior parietal lobe | 0.0161 | 0.0707 | 0.8202 | 0.9266 | 0.0065 | 0.0048 | 0.1795 | 0.2434 |
| | Precuneus | 0.0574 | 0.0929 | 0.5365 | 0.8759 | 0.0110 | 0.0055 | **0.0446** | 0.0971 |
| | Supramarginal gyrus | −0.0736 | 0.0658 | 0.2639 | 0.8759 | 0.0070 | 0.0040 | 0.0852 | 0.1443 |
| | Angular gyrus | 0.1239 | 0.0781 | 0.1132 | 0.6791 | 0.0075 | 0.0051 | 0.1376 | 0.1908 |
| | Entorhinal cortex | −0.0182 | 0.0187 | 0.3316 | 0.8759 | 0.0052 | 0.0019 | **0.0060** | **0.0416** |
| | Parahippocampal gyrus | −0.0180 | 0.0243 | 0.4588 | 0.8759 | 0.0058 | 0.0021 | **0.0054** | **0.0416** |
| | Parietal white matter | 0.3078 | 0.2397 | 0.1996 | 0.8118 | 0.0150 | 0.0172 | 0.3858 | 0.4441 |
| **Temporal** | Temporal pole | −0.0493 | 0.0759 | 0.5166 | 0.8759 | 0.0106 | 0.0052 | **0.0417** | 0.0942 |
| | Superior temporal gyrus | 0.0202 | 0.0584 | 0.7295 | 0.8852 | 0.0068 | 0.0035 | 0.0524 | 0.1102 |
| | Middle temporal gyrus | 0.1419 | 0.0945 | 0.1336 | 0.6791 | 0.0196 | 0.0073 | **0.0074** | **0.0416** |
| | Inferior temporal gyrus | −0.0217 | 0.0698 | 0.7562 | 0.8871 | 0.0132 | 0.0061 | **0.0319** | 0.0847 |
| | Fusiform gyrus | 0.0909 | 0.0643 | 0.1581 | 0.7417 | 0.0035 | 0.0040 | 0.3823 | 0.4441 |
| | Temporal white matter | 0.0450 | 0.2442 | 0.8540 | 0.9471 | 0.0323 | 0.0155 | **0.0370** | 0.0930 |
| **Occipital** | Occipital pole | 0.0544 | 0.0399 | 0.1732 | 0.7547 | 0.0027 | 0.0024 | 0.2688 | 0.3488 |
| | Superior occipital gyrus | −0.0161 | 0.0393 | 0.6814 | 0.8852 | 0.0059 | 0.0019 | **0.0016** | **0.0416** |
| | Middle occipital gyrus | 0.0171 | 0.0516 | 0.7401 | 0.8852 | 0.0089 | 0.0033 | **0.0070** | **0.0416** |
| | Inferior occipital gyrus | −0.0177 | 0.0492 | 0.7193 | 0.8852 | 0.0081 | 0.0037 | **0.0313** | 0.0847 |
| | Occipital fusiform gyrus | 0.0990 | 0.0357 | **0.0057** | 0.3478 | 0.0085 | 0.0031 | **0.0063** | **0.0416** |
| | Medial occipital | 0.1158 | 0.1214 | 0.3404 | 0.8759 | 0.0207 | 0.0079 | **0.0085** | **0.0416** |
| | Occipital white matter | 0.3742 | 0.1572 | **0.0176** | 0.3987 | −0.0045 | 0.0075 | 0.5469 | 0.5752 |
| **Cingulate** | Anterior cingulate gyrus | 0.0211 | 0.0463 | 0.6493 | 0.8852 | 0.0070 | 0.0023 | **0.0028** | **0.0416** |
| | Middle cingulate gyrus | −0.0007 | 0.0375 | 0.9844 | 0.9844 | 0.0057 | 0.0022 | **0.0081** | **0.0416** |
| | Posterior cingulate gyrus | 0.0197 | 0.0322 | 0.5409 | 0.8759 | 0.0041 | 0.0020 | **0.0381** | 0.0930 |
| **Basal ganglia and related** | Accumbens area | 0.0039 | 0.0049 | 0.4337 | 0.8759 | −0.0001 | 0.0004 | 0.6908 | 0.7142 |
| | Basal forebrain | 0.0020 | 0.0058 | 0.7364 | 0.8852 | 0.0023 | 0.0010 | **0.0155** | 0.0557 |
| | Amygdala | −0.0032 | 0.0082 | 0.6979 | 0.8852 | 0.0015 | 0.0008 | 0.0627 | 0.1234 |
| | Hippocampus | −0.0190 | 0.0262 | 0.4680 | 0.8759 | 0.0020 | 0.0019 | 0.2954 | 0.3678 |

**Table 2 (continued) | Associations between skeletal muscle oxidative capacity and brain volumes via MRI (n = 649)**

| | | Cross-sectional associations | | | | Longitudinal associations | | | |
|---|---|---|---|---|---|---|---|---|---|
| | | β | SE | *p*-value | FDR-adj. *p* | β | SE | *p*-value | FDR-adj. *p* |
| | Caudate | −0.0043 | 0.0363 | 0.9068 | 0.9704 | 0.0003 | 0.0022 | 0.9066 | 0.9217 |
| | Putamen | −0.0031 | 0.0345 | 0.9286 | 0.9766 | 0.0041 | 0.0024 | 0.0841 | 0.1443 |
| | Globus pallidus | 0.0114 | 0.0113 | 0.3107 | 0.8759 | 0.0013 | 0.0007 | 0.0666 | 0.1270 |
| | Thalamus | 0.0180 | 0.0395 | 0.6478 | 0.8852 | 0.0054 | 0.0025 | **0.0315** | 0.0847 |
| | Anterior limb of the internal capsule | 0.0201 | 0.0224 | 0.3710 | 0.8759 | −0.0001 | 0.0013 | 0.9457 | 0.9457 |
| | Posterior limb of the internal capsule | 0.0062 | 0.0175 | 0.7223 | 0.8852 | 0.0024 | 0.0014 | 0.0817 | 0.1443 |
| | Fornix | −0.0002 | 0.0051 | 0.9723 | 0.9844 | 0.0008 | 0.0005 | 0.1114 | 0.1699 |
| **CC** | Corpus callosum | −0.0421 | 0.0562 | 0.4542 | 0.8759 | 0.0026 | 0.0022 | 0.2397 | 0.3179 |
| **Cerebellum** | Cerebellum exterior gray matter | −0.1891 | 0.3136 | 0.5468 | 0.8759 | 0.0138 | 0.0169 | 0.4149 | 0.4601 |
| | Cerebellum white matter | 0.0156 | 0.1046 | 0.8818 | 0.9605 | 0.0093 | 0.0042 | **0.0255** | 0.0778 |

Note: Statistical results are from linear mixed effects models with significance at two-sided $p < 0.05$. Bold number indicates $p$ or FDR-ad. $p < 0.05$. $k_{PCr}$ and SPARE scores are standardized to Z scores. Magnetic resonance imaging (MRI) volumes are on the original scale in $cm^3$ for interpretation purposes.

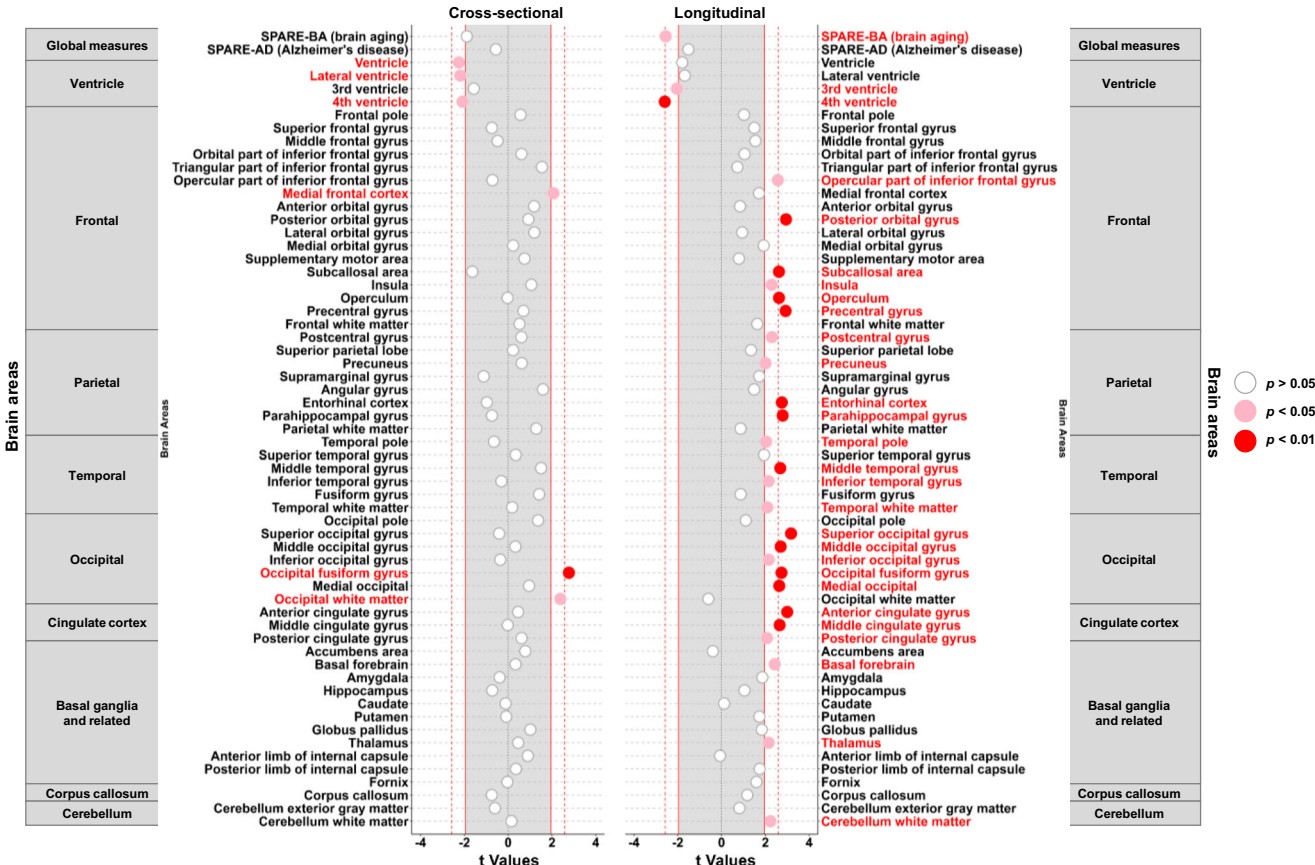

**Fig. 2 | Dot plots for associations between skeletal muscle oxidative capacity and brain volumetric measures via magnetic resonance imaging (MRI) (n = 649).** Legend: Statistical results are from linear mixed effects models with significance at two-sided *p*-values. Exact *p*-values are the same as in Table 2. Data points beyond red solid lines are at $p < 0.05$ and beyond red dotted lines are at $p < 0.01$ and also survived FDR-$p < 0.05$. Source data are provided as a Source Data file.

hemispheres and is important for bimanual coordination[23], as well as communication essential to the coordination of lateralized cognitive functions. These specific tracts are centered around the cingulate cortex, which is important for multiple functions, such as emotion, cognition, and motor control[24], and key in successful cognitive aging[25]. The cingulate cortex is adjacent to the corpus callosum which is extended by the cingulum. The superior longitudinal fasciculus runs above the cingulum and the cingulate cortex also connects frontal, parietal, temporal, and occipital lobes. Notably, we did not find significant associations with white matter atrophy in the corpus callosum. DTI measures may be more sensitive to white matter changes than volumetric measures[26]. We also examined gray matter microstructure

**Table 3 | Associations between skeletal muscle oxidative capacity and white matter microstructural measures of mean diffusivity and fractional anisotropy via DTI (*n* = 639)**

| | | Fractional Anisotropy (higher value: higher integrity) | | | | | |
| --- | --- | --- | --- | --- | --- | --- | --- |
| | | Cross-sectional associations | | | Longitudinal associations | | |
| | | β | SE | *p*-value | β | SE | *p*-value |
| **Commissural fibers** | Genu of corpus callosum | 0.0480 | 0.0329 | 0.1448 | 0.0010 | 0.0044 | 0.8118 |
| | Body of corpus callosum | 0.0298 | 0.0361 | 0.4092 | 0.0072 | 0.0036 | **0.0458** |
| | Splenium of corpus callosum | 0.0816 | 0.0401 | **0.0426** | 0.0068 | 0.0037 | 0.0648 |
| **Association fibers** | External capsule | 0.0080 | 0.0309 | 0.7956 | 0.0023 | 0.0051 | 0.6496 |
| | Uncinate fasciculus | −0.0297 | 0.0411 | 0.4706 | −0.0022 | 0.0049 | 0.6575 |
| | Superior longitudinal fasciculus | −0.0159 | 0.0382 | 0.6780 | 0.0077 | 0.0038 | **0.0442** |
| | Inferior longitudinal fasciculus | 0.0002 | 0.0393 | 0.9955 | 0.0024 | 0.0042 | 0.5577 |
| | Superior fronto-occipital fasciculus | 0.0208 | 0.0385 | 0.5891 | 0.0056 | 0.0052 | 0.2861 |
| | Inferior fronto-occipital fasciculus | 0.0057 | 0.0390 | 0.8829 | 0.0071 | 0.0054 | 0.1903 |
| | Cingulate part of the cingulum | 0.0319 | 0.0390 | 0.4143 | 0.0067 | 0.0036 | 0.0644 |
| | Hippocampal part of the cingulum | 0.0135 | 0.0381 | 0.7231 | 0.0108 | 0.0059 | 0.0648 |
| | Fornix (column and body) | 0.0413 | 0.0338 | 0.2223 | 0.0025 | 0.0029 | 0.3955 |
| | Fornix (cres) Stria terminalis | 0.0549 | 0.0328 | 0.0943 | 0.0076 | 0.0040 | 0.0578 |
| **Projection fibers** | Anterior limb of the internal capsule | 0.0070 | 0.0348 | 0.8403 | 0.0037 | 0.0050 | 0.4596 |
| | Posterior limb of the internal capsule | −0.0185 | 0.0398 | 0.6428 | 0.0063 | 0.0061 | 0.3071 |
| | Anterior corona radiata | −0.0096 | 0.0321 | 0.7642 | −0.0010 | 0.0031 | 0.7402 |
| | Superior corona radiata | −0.0337 | 0.0397 | 0.3962 | 0.0000 | 0.0036 | 0.9954 |
| | Posterior corona radiata | −0.0445 | 0.0409 | 0.2778 | 0.0024 | 0.0039 | 0.5429 |
| **Cerebellum** | Cerebellar peduncle | 0.0481 | 0.0361 | 0.1833 | 0.0017 | 0.0080 | 0.8327 |
| | | Mean Diffusivity (lower value: higher integrity) | | | | | |
| | | Cross-sectional associations | | | Longitudinal associations | | |
| | | β | SE | *p*-value | β | SE | *p*-value |
| **Commissural fibers** | Genu of corpus callosum | −0.0563 | 0.0334 | 0.0921 | −0.0021 | 0.0046 | 0.6508 |
| | Body of corpus callosum | −0.0401 | 0.0335 | 0.2315 | −0.0052 | 0.0040 | 0.1921 |
| | Splenium of corpus callosum | −0.0842 | 0.0385 | **0.0292** | −0.0052 | 0.0038 | 0.1714 |
| **Association fibers** | External capsule | −0.0032 | 0.0292 | 0.9117 | −0.0014 | 0.0036 | 0.6946 |
| | Uncinate fasciculus | 0.0016 | 0.0356 | 0.9649 | 0.0057 | 0.0046 | 0.2077 |
| | Superior longitudinal fasciculus | 0.0083 | 0.0337 | 0.8053 | −0.0040 | 0.0044 | 0.3552 |
| | Inferior longitudinal fasciculus | −0.0190 | 0.0375 | 0.6136 | −0.0044 | 0.0059 | 0.4553 |
| | Superior fronto-occipital fasciculus | −0.0121 | 0.0335 | 0.7173 | −0.0002 | 0.0049 | 0.9591 |
| | Inferior fronto-occipital fasciculus | −0.0150 | 0.0323 | 0.6419 | −0.0052 | 0.0040 | 0.1997 |
| | Cingulate part of the cingulum | −0.0653 | 0.0356 | 0.0670 | −0.0042 | 0.0054 | 0.4363 |
| | Hippocampal part of the cingulum | −0.0837 | 0.0390 | **0.0325** | −0.0068 | 0.0059 | 0.2522 |
| | Fornix (column and body) | −0.0423 | 0.0332 | 0.2038 | 0.0014 | 0.0027 | 0.5944 |
| | Fornix (cres) Stria terminalis | −0.0523 | 0.0339 | 0.1228 | −0.0012 | 0.0038 | 0.7596 |
| **Projection fibers** | Anterior limb of the internal capsule | −0.0204 | 0.0328 | 0.5340 | −0.0002 | 0.0046 | 0.9678 |
| | Posterior limb of the internal capsule | −0.0249 | 0.0336 | 0.4584 | −0.0105 | 0.0054 | 0.0522 |
| | Anterior corona radiata | 0.0035 | 0.0334 | 0.9170 | 0.0002 | 0.0039 | 0.9493 |
| | Superior corona radiata | −0.0121 | 0.0332 | 0.7146 | −0.0020 | 0.0040 | 0.6185 |
| | Posterior corona radiata | 0.0000 | 0.0378 | 0.9998 | −0.0034 | 0.0038 | 0.3669 |
| **Cerebellum** | Cerebellar peduncle | −0.0493 | 0.0369 | 0.1819 | −0.0017 | 0.0071 | 0.8095 |

Note: Statistical results are from linear mixed effects models with significance at two-sided $p < 0.05$. Bold number indicates $p < 0.05$. $k_{PCr}$ and diffusion tensor imaging (DTI) values are standardized to Z scores.

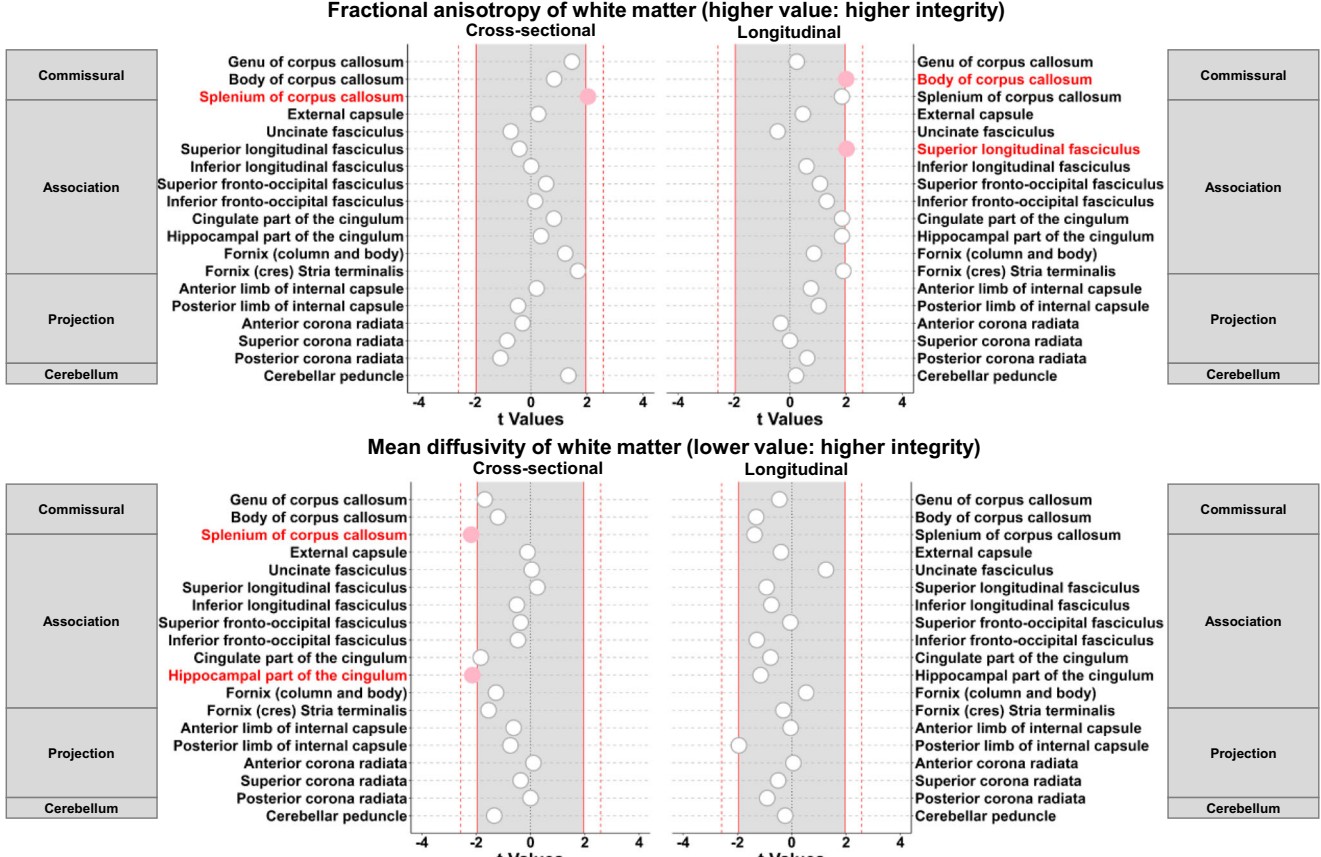

**Fig. 3 | Dot plots for associations between skeletal muscle oxidative capacity and white matter diffusion tensor imaging (DTI) measures of fractional anisotropy (FA) and mean diffusivity (MD) ($n$ = 639).** Legend: Statistical results are from linear mixed effects models with significance at two-sided $p < 0.05$. Exact $p$-values are the same as in Table 3. Data points beyond red solid lines are at $p < 0.05$ and beyond red dotted lines are at $p < 0.01$. Source data are provided as a Source Data file.

but did not find any significant longitudinal associations with either FA or MD. Why associations were found in gray matter atrophy and not in gray matter microstructure is unclear.

Our findings highlight the importance of mitochondrial health in brain aging and underline that the search for a mechanism explaining this connection warrants further investigation and may help find new intervention targets. One potential mechanism is that muscle mitochondrial function indicates general mitochondrial health and that muscle mitochondria can be considered a proxy measure of mitochondrial health across multiple tissues, including the brain. Another possibility is that the measure of oxidative capacity captures general muscle health and that positive signaling through soluble molecules and/or microvesicles may act in neurotrophic signaling that promotes brain health[27]. While skeletal muscle oxidative capacity is related to fitness, the longitudinal associations between skeletal muscle oxidative capacity and brain atrophy were independent of concurrent fitness levels. Longitudinal associations with microstructural change persisted after accounting for the fitness measure of 400-meter walk time but were attenuated after adjusting for $VO_2$ max. This attenuation is not surprising as fitness and vascular factors are strongly associated with white matter microstructure captured by DTI[28–31]. It is also possible that the collinearity between $k_{PCr}$ and $VO_2$ max contributed to the attenuation of these associations. Because of the observational nature of this study, the detected longitudinal associations may shed light on but do not prove a causal relationship. In addition, we cannot exclude that higher skeletal muscle oxidative capacity

reflects in part the lifetime history of exercise and physical activity which may affect several aspects of brain health but may not be fully captured by the assessment of current fitness levels.

Regardless of the mechanisms, an emerging question is why certain brain areas are correlated with muscle mitochondria oxidative capacity. Which area of the brain has the highest mitochondrial component? What areas are more energy demanding at rest and during increased activity, therefore being more sensitive to mitochondrial dysfunction? Initial brain MRS studies have shown that mitochondrial function is negatively associated with increasing age and some metabolites show sex differences[32,33]. Future studies are warranted to directly assess in vivo mitochondrial function in the brain via MR spectroscopy and examine its relationship with skeletal muscle oxidative capacity, which can further elucidate underlying mechanisms of mitochondria in the aging CNS and neurodegeneration.

This study has limitations. First, the study sample tends to be healthier than the general population due to their voluntary participation in the BLSA and eligibility to participate in the MRI studies of the brain and the skeletal muscle. Second, this longitudinal study does not address causation due to the observational study design. This study has several strengths. First, MR spectroscopy-based skeletal muscle oxidative capacity involving an exercise protocol provides an in vivo, non-invasive assessment of mitochondrial function in a dynamic process. Second, the two neuroimaging modalities of MRI and DTI provide complementary information on gray matter and white matter macro and microstructure, advancing our understanding of the brain spatial distribution with muscle mitochondrial function. Third, up to 11 years of repeated neuroimaging measures allow us to investigate

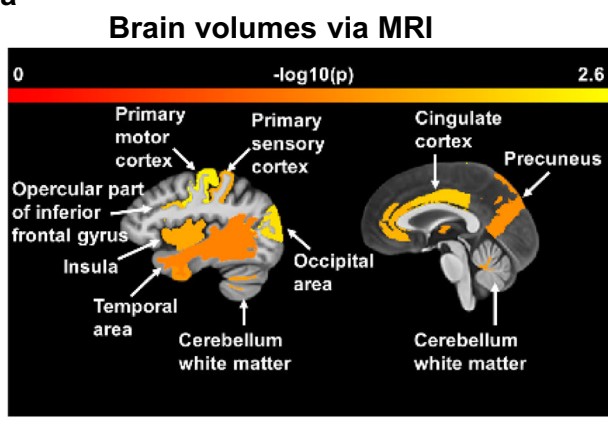

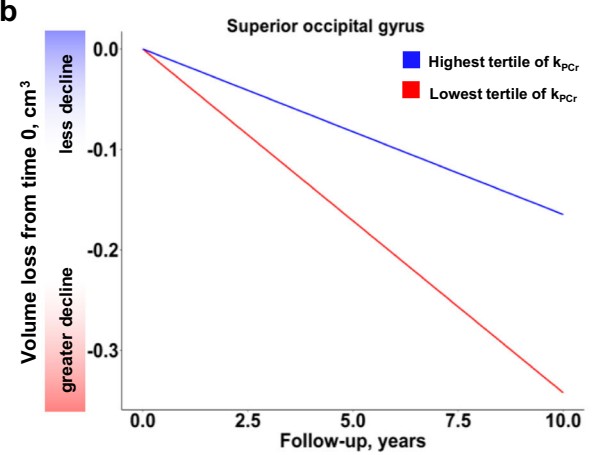

**Fig. 4 | Brain areas that showed significant longitudinal associations with skeletal muscle oxidative capacity (parts a and c) and estimated trajectories of top significant volumetric (n = 649) (part b) and microstructural (n = 639) (part d) measures at higher and lower levels of skeletal muscle oxidative capacity.**

Legend: Statistical results are from linear mixed effects models with significance at two-sided $p < 0.05$. Time 0 is the first assessment of skeletal muscle oxidative capacity concurrent with brain magnetic resonance imaging (MRI) scans. Source data are provided as a Source Data file.

longitudinal brain changes within aging. Last but not least, the sensitivity analysis of adjustment for total brain volume confirms the strength of the brain regional associations which are less likely to be affected by the overall brain atrophy.

In conclusion, findings from this longitudinal study with up to over a decade-long neuroimaging measures provide firsthand evidence of the key role of mitochondrial health in aging CNS and reveal spatial distributions. Mitochondria health assessed in the skeletal muscle is associated with overall and regional brain atrophy as well as preserved microstructural integrity of connecting white matter tracts. These brain areas are important for cognition, motor function, and sensorimotor integration and may be sensitive to hypoperfusion. Future research is needed to understand whether the longitudinal decline in mitochondrial function is related to longitudinal changes in aging CNS and neurodegeneration and to further elucidate underlying neural substrates.

## Methods
### Study population
Participants were drawn from the Baltimore Longitudinal Study of Aging (BLSA), a longitudinal study with continuous enrollment since 1958[34,35]. At enrollment, participants were free of cognitive and functional impairment and chronic conditions. At each visit, participants

undergo health, cognitive, and functional assessments at the National Institute on Aging Clinical Research Unit over 3 days. Participants were followed up every 4 years if younger than age 60, every 2 years between ages 60–79, and every year if age 80 or older. Information on sex (i.e., men, women) was obtained by self-report at the BLSA visit.

In this study, we used the first assessment of skeletal muscle mitochondrial function to examine the relationship with longitudinal changes in brain structure between 2008 and 2020. A sample of 649 participants was included for analysis. The BLSA protocol was approved by the Institutional Review Board of the National Institutes of Health. All participants provided written informed consent at each study visit.

### Skeletal muscle mitochondrial function
In vivo mitochondrial function was measured as maximal oxidative capacity from the quadriceps muscles using $^{31}P$ magnetic resonance (MR) spectroscopy on a 3 T Philips Achieva MR scanner (Philips, Best, The Netherlands) since April 2013 in the BLSA. Participants were instructed to perform a fast and intense ballistic knee extension exercise while positioned supine in the bore of the scanner to deplete phosphocreatine (PCr) in the quadriceps muscles with minimal acidification. $^{31}P$-magnetic resonance spectroscopy (MRS) spectra with a pulse-acquire sequence before, during, and after exercise were

acquired using a 10 cm, flat surface coil (PulseTeq, Surrey, United Kingdom) secured over the vastus lateralis muscle of the left thigh. The exercise was performed until a reduction in PCr peak height between 33% and 66% was achieved. The total duration of MRS data acquisition was 7.5 min. Spectral processing was performed using jMRUI (version 5.2). Skeletal muscle mitochondrial function or maximal oxidative capacity was quantified as the post-exercise PCr recovery rate constant, $k_{PCr}$. A higher value of $k_{PCr}$ indicates higher mitochondrial function. Participants may not be able to complete the protocol due to health-related reasons, such as lower limb pain, inability to coordinate to perform a smooth, fast kicking motion, and impaired hearing impacting instruction receival and subsequent comprehension. Participants with at least 33% PCr peak height or area depletion during the exercise phase were included in the analysis.

The exercise test used in this study is based on the theory of the creatine phosphate shuttle, a system that facilitates the transport of high-energy phosphate from muscle cell mitochondria to myofibrils[36]. At rest, ATP levels are high leading to the conversion of creatine to phosphocreatine. During exercise in the magnet, phosphocreatine breaks down to creatine and inorganic phosphate, which are monitored by the MRI coil. At the end of the exercise, mitochondria provide ATP to rephosphorylate creatine into creatine phosphate. The rate of rephosphorylation mostly depends on mitochondrial oxidative capacity and therefore this rate monitored by MRI is a good proxy measure of mitochondrial oxidative capacity[37]. This protocol has been used in other studies[38–40]. Skeletal muscle oxidative capacity is associated with mitochondrial DNA copy number from blood samples[41].

## Brain MRI

Imaging data were acquired on 3 comparable 3 T Philips Achieva scanners at the Kennedy Krieger Institute (KKI) or the National Institute on Aging (NIA) in Baltimore, Maryland. Magnetization-prepared rapid gradient-echo (MPRAGE: repetition time = 6.8 ms, echo time = 3.2 ms, flip angle = 8°, image matrix = 256 × 256 × 170, and voxel size = 1 × 1 × 1.2 mm³) scans were acquired in the sagittal plane. MPRAGE scan anatomical labels and regional brain volumes were computed through Multi-atlas region Segmentation using Ensembles (MUSE) of registration algorithms and parameters[42]. The MUSE methodology is shown to be accurate and robust and has been used in various datasets[42,43]. In addition, a detailed and validated harmonization approach has been incorporated into the analysis of structural MRI scans[43]. Longitudinal measures of regional brain volumes by MRI have good stability and consistency with a previously reported intraclass correlation between 0.89 and 0.99[44].

In this study, we focused on longitudinal changes in both global and regional brain volumetric measures, including ventricular volumes, regional gray matter volumes, and regional white matter volumes. Gray matter ROIs include frontal, parietal, temporal, occipital, cingulate, basal ganglia and related, and cerebellum regions. White matter ROIs included white matter from frontal, parietal, temporal, and occipital lobes as well as cerebellum white matter. We also examined previously validated brain aging (SPARE-BA) and AD-like atrophy (SPARE-AD) indices derived from brain MRI[45]. A greater increase in a SPARE score over time indicates accelerated brain atrophy in a certain spatial pattern. Specifically, an increase in the SPARE-BA score indicates patterns of brain atrophy associated with advanced brain aging, whereas an increase in SPARE-AD indicates a more AD-like pattern of brain atrophy.

## Brain DTI

The diffusion tensor imaging (DTI) acquisition protocol was identical for the two KKI scanners: number of gradients = 32, number of b0 images = 1, max b-factor = 700 s/mm², TR/TE = 6801/75 msec, number of slices = 65, voxel size = 0.83 × 0.83 × 2.2 mm, reconstruction matrix = 256 × 256, acquisition matrix = 96 × 95, field of view = 212 × 212 mm, flip angle = 90°.

The DTI acquisition protocol for the NIA scanner was as follows: number of gradients = 32, number of b0 images = 1, max b-factor = 700 s/mm², repetition time/echo time (TR/TE) = 7454/75 msec, number of slices = 70, voxel size = 0.81 × 0.81 × 2.2 mm, reconstruction matrix = 320 × 320, acquisition matrix = 116 × 115, field of view = 260 × 260 mm, flip angle = 90°. Each participant received 2 DTI scans. Two b0 images were averaged in k-space for each DTI acquisition. Two separate DTI acquisitions with the number of signal average (NSA) = 1 were obtained and combined offline for an effective NSA = 2 to improve signal-to-noise ratio[46]. DTI processing followed standard practice for tensor fitting and quality assessment explained in detail in earlier publications[46,47]. To compensate for eddy current effects and physiological motion, individual diffusion-weighted volumes were affine co-registered to a minimally weighted (b0) target. Gradient tables were corrected for the identified rotational component using finite strain[48]. Each diffusion-weighted image was normalized by its own reference image before tensor fitting to combine the 2 DTI sessions with different and unknown intensity normalization constants. Quality control (QC) was performed to remove scans with excessive motion or images with globally high diffusion measure bias after reviewing distributions of QC summary statistics[46]. Longitudinal measures of brain DTI measures of FA and MD show average intraclass correlations of 0.76 and 0.66, respectively, and within-scanner versus between-scanner intraclass correlations were comparable for the measures at 3 T[49].

In this study, we examined longitudinal changes in fractional anisotropy (FA) and mean diffusivity (MD) of both gray matter and white matter. Metrics of DTI, such as FA and MD, are widely used to quantify water molecules in white matter fiber tracks and can also provide quantitative measures of gray matter microstructure because of unconstrained water molecules in cerebrospinal fluid and adjacent areas such as cerebral cortex[50]. FA describes the degree of anisotropy of water molecules and higher FA indicates higher microstructural integrity. MD describes the magnitude of water diffusion within tissues considering all directions and higher MD indicates lower microstructural integrity. For white matter, we also explored radial diffusivity (RD) and axial diffusivity (AD). RD describes the magnitude of water diffusion perpendicular to fiber tracts. AD describes the magnitude of water diffusion parallel to fiber tracts. Higher values of RD and AD indicate lower microstructural integrity.

For microstructure in gray matter, we examined the same ROIs as volumetric analysis. To segment gray matter, we used multi-atlas registration with the BrainCOLOR protocol using 35 atlases labeled manually from NeuroMorphometrics[51]. Labels of ROIs obtained from the T1 image were affine registered to the diffusion image and used to extract region-specific average MD and FA measures. For white matter tracts, we examined commissural fibers (corpus callosum), association tracts (external capsule, uncinate fasciculus, superior longitudinal fasciculus, inferior longitudinal fasciculus, superior fronto-occipital fasciculus, inferior fronto-occipital fasciculus, cingulum, fornix), projection tracts (internal capsule and corona radiata), and a tract through the cerebellum (cerebellar peduncle). For white matter segmentation, we used the Eve White Matter atlas with white matter labels from multi-atlas segmentation and an FA-mapped MRI[51,52]. White matter labels and white matter segmentation were intersected, and resulting labels were iteratively grown to fill the remaining white matter space from the multi-atlas labels. The white matter labels of ROIs obtained from the T1 image were affine registered to the FA/MD/AD/RD image and used to extract average FA/MD/AD/RD measures for each area of interest.

## Statistical analysis

To examine the relationship between skeletal muscle mitochondrial function and longitudinal changes in brain structure, we used linear mixed effect (LME) regression models. For neuroimaging measures, we checked for outliers using LME regression with age and $age^2$ as predictors. Observations with residual values >5 standard deviations

were removed from the analysis (<0.8%). The first concurrent assessment visit of skeletal muscle oxidative capacity and brain imaging scans was determined as the index visit. The time interval for data points before the index visit was modeled as −1 year, −2 years, etc., whereas data points after the index visit were modeled as +1 year, +2 years, etc. Models were adjusted for age, sex, race, physical activity, and PCr depletion, and adjusted for intracranial volume (ICV) estimated at age 70 for brain atrophy measures. Adjustment for a single baseline measure of ICV was to avoid any effect of potential changes on ICV measures over time[53]. Age 70 was around the average age of the BLSA population at the baseline MRI. Physical activity was measured by self-report and participants were classified as inactive or low, moderate, or very high activity[54,55]. Fixed effects included covariates and their interactions with the time interval. Random effects included intercept and interval. We reported both cross-sectional associations at the index visit and longitudinal associations with brain structural changes up to 11 years. $k_{PCr}$ values were standardized to Z scores due to small values. DTI and SPARE scores were unitless and standardized to Z scores. MRI values were on the original scale in $cm^3$ for interpretation purposes.

To understand whether global atrophy affected the association with regional brain atrophy, in a sensitivity analysis we adjusted for total brain volume at the first MRI visit instead of intracranial volume. To understand how cognitive impairment affected the longitudinal associations, we repeated the analyses by removing data points at and after the symptom onset of cognitive impairment. Because skeletal muscle oxidative capacity is associated with fitness[56–58], we performed sensitivity analyses by adjusting the longitudinal analysis for 400 meter walk time from the Long-Distance Corridor Walk or $VO_2$ max from a maximal exercise treadmill test, which were both considered reliable tests of fitness. Of the 649 participants in the main analysis, 627 had data on 400-meter walk time and 586 had data on $VO_2$ max. Because physical activity and fitness were highly correlated, models adjusting for fitness did not include physical activity. We performed additional sensitivity analyses by adjusting for apolipoprotein E ε4 (APOE ε4) carrier status and by limiting the analysis to participants aged 60 years or older. We also explored sex differences by adding an interaction term with sex to the LME models. In this exploratory study, significance was set at a two-tailed $p < 0.05$. We also reported False Discovery Rate (FDR)-adjusted $p$-values to correct for multiple testing. All LME models were fitted using the "nlme" package of RStudio version 4.3.1 (Boston, MA) (Supplementary Data 1).

### Inclusion and ethics statements
This research follows the guidelines of inclusion and ethics by Nature Communications. All authors have fulfilled the authorship criteria.

### Reporting summary
Further information on research design is available in the Nature Portfolio Reporting Summary linked to this article.

## Data availability
Data analyzed in this study are available upon request by proposal submission via the BLSA website portal (https://www.blsa.nih.gov/how-apply). All requests to access the BLSA datasets are reviewed by the BLSA Data Sharing Proposal Review Committee and are also subject to approval from the NIH Institutional Review Board. Source data are provided with this paper.

## Code availability
We provided the script used for the linear mixed effects modeling in Supplementary Data 1. Additional codes used in this study are available upon request.

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

## Acknowledgements

This study was supported in part by the Intramural Research Program of the National Institute on Aging, Baltimore, Maryland. We would like to acknowledge Dr. Kurt Schilling and Mr. Sridhar Kandala for their assistance in brain imaging figures, and Ms. Lisa M. Hartnell for producing the featured image. Also, we would like to acknowledge the study participants and staff of the Baltimore Longitudinal Study of Aging for their participation and dedication.

## Author contributions

Study concept and design: Q.T., E.E.G., S.M.R., L.F.; Data acquisition, analysis, or interpretation of data: Q.T., E.E.G., C.D., B.A.L., S.M.R., L.F.; Drafted the manuscript: Q.T. .and E.E.G.; all authors (Q.T., E.E.G., C.D., B.A.L., S.M.R., L.F.) substantially revised it, have approved the submitted version and have agreed to both to be personally accountable for the authors' own contributions and to ensure that questions related to the accuracy or integrity of any part of the work are appropriately investigated, resolved, and the resolution documented in the literature.

## Funding

## Competing interests

The Authors declare no competing interests.
