## [Transparent Peer Review file · Nature Communications]

Higher skeletal muscle mitochondrial oxidative capacity is associated with preserved brain structure up over a decade

Corresponding Author: Dr Qu Tian

Version 0:

Reviewer comments:

Reviewer #1

(Remarks to the Author)

The submitted manuscript by Tian et al. examined the association between in vivo skeletal muscle mitochondrial function and multiple MRI measures cross-sectionally and longitudinally utilizing data from the Baltimore Longitudinal Study of Aging (BLSA). MRI/DTI outcomes included global and regional brain volumes, as well as white/gray matter microstructure (e.g., fractional anisotropy (FA), mean diffusivity (MD), radial diffusivity (RD), and axial diffusivity (AD)). The study found associations between higher skeletal muscle mitochondrial function and less atrophy over time in 13 structural ROIs after correction for multiple testing.

The manuscript addresses an important area of research, as much work remains to be done to understand the role of mitochondrial function in brain aging. The paper is well-written, includes high-quality longitudinal imaging data from a diverse sample, and employs appropriate statistical methods. However, certain areas could be improved to enhance the clarity and impact of the work.

Major comments:

1. The authors state that skeletal muscle mitochondria oxidative capacity is a proxy for overall mitochondrial function, including in the brain. However, no information is provided to support this statement in the introduction. Skeletal muscle mitochondria oxidative capacity can simply reflect fitness, which is related to better neurological outcomes, and adjustment for self-reported physical activity may only account for a fraction of fitness. Can the authors provide additional information to support this statement? For example, are there studies correlating quadriceps and brain 31P-MRS data, or other experimental models? Expanding these points in the discussion would be appropriate to contextualize the results.

2. It would be helpful if the authors could provide a flowchart indicating how they arrived at the final sample of 649 participants. No information is provided on the number of participants who were excluded. Additionally, a table of participants' characteristics presenting the original, cross-sectional, and longitudinal samples would help identify selection bias.

3. Were there any differences in the results by sex?

4. It would be interesting to include global DTI markers of FA and MD in total white matter — there may be additional signal than looking at individual tracts.

5. It is somewhat unusual to use DTI metrics in gray matter. FA and MD may not be as relevant in gray matter, given the more restricted and heterogeneous diffusion of water in cell bodies, as opposed to white matter axons. Can the authors provide a rationale for including gray matter FA and MD as outcomes of interest?

6. I recommend toning down conclusions on findings related to DTI outcomes, as these did not pass correction for multiple testing. Currently, the discussion suggests these are “robust”, and “strongly related” to mitochondrial function.

Minor comments:

7. The brain MRI methods section includes a paragraph in quotation, presumably because the same text has been used before. I would recommend either citing the source at the end of the paragraph.
8. Figure 3 is mentioned before Figure 2 in the text, so it needs to be reorganized in the manuscript.
9. Figure 1 does not show global measurements (SPARE-BA, SPARE-AD).
10. Figures 1 and 2 should clearly indicate that bold red data points correspond to FDR-corrected results in the legend.
11. Tables 2 and 3 could be added to the supplementary material as the same information is presented in Figures 1 and 2.
12. Why do global and ventricular outcomes not have FDR-adjusted p-values in Table 2?
13. Supplementary Table 1 has repeated headings, radial diffusivity is missing.

Co-reviewed by Claudia Satizabal & Janette Vazquez

Reviewer #2

(Remarks to the Author)

The authors used data from the Baltimore Longitudinal Study of Aging (BLSA) to examine the association between in vivo skeletal muscle maximal oxidative capacity after exercise and longitudinal changes in brain structure assessed using imaging. While the study results represent an important contribution, the authors could contextualize better the mitochondria biology measure they are using and how it relates to brain function (see specific comments below).

Introduction

- Line 51. "Mitochondrial function assessed in the skeletal muscle and blood predicts future cognitive impairment and dementia and is associated with cognitive function and biomarkers of AD and neurodegeneration (Tian, Bilgel, et al., 2023; Tian et al., 2024; Tian, Zweibaum, et al., 2023; Yang et al., 2021; Zhang et al., 2023)." Mitochondria have several functions, could you be specific about the mitochondrial outcomes measured in these studies?
- Line 55, what was the mitochondrial function assessed?
- Line 60 Do you have references to support that mitochondrial biology in skeletal muscle reflects brain mitochondrial biology? Did any study look at the correlation between mitochondrial measures in muscle vs brain?
- Line 64, which function specifically? Could you introduce the measure you are using (skeletal muscle mitochondrial oxidative capacity) and explain which function it reflects? Why more of it after exercise is better?

Method

- Has that measure of maximal oxidative capacity been used by other study? Does it correlate with other mitochondrial markers?

Results

- The individual datapoints are not shown, some illustrative scatterplots could be used for transparency.

Discussion

- 274-277: is there any prior evidence suggesting that skeletal muscle bioenergetics measures correlate with brain bioenergetics measures?
- Is it plausible that healthier individuals have better skeletal muscle bioenergetics and better brain health without the two being causality related?

Reviewer #3

(Remarks to the Author)

This study leverages a large and well characterized cohort, the Baltimore Longitudinal Study of Aging, to examine the interesting and important relationship between skeletal muscle mitochondrial function and brain imaging outcomes. There are key strengths of the study including the inclusion of multiple imaging sequences and functional mitochondrial assessments. This type of study is needed and overdue, is novel, and contributes to the literature. However, there are several clarifications and additions that are needed to strengthen the manuscript.

There was no clear formal hypothesis presented in the introduction. What did the investigators expect to find?

For cross-sectional measures, what was the mean length (and range) of durations between when the MRI scan was obtained and when the MR muscle functional mitochondrial assessment was performed?

Based on the study design, there are differences in the length of follow up MRI scans based on age bins. Please clarify within the manuscript how many participants fell into each bin (younger than 60, 60-79, 80+)? Given that the first assessment of muscle mitochondrial function was used, there would have been varying frequency of MRI scans for each person during the longitudinal follow up period. How do the investigators feel this may affect the results (especially cross sectional findings)?

For brain volume measures, did investigators assess hemisphere-specific differences (L vs R)? In addition to differences in cross sectional volumes between hemispheres in some brain regions, the rate of longitudinal atrophy can differ between hemispheres. Was handedness assessed?

Given that multiple MRI scanners were used, details should be added for structural QC procedures, phantom scans, etc. that were run to ensure reproducibility across scanners. The only QC that was mentioned was related to the DTI.

Results section: please include statistics (i.e. p values, regression coefficients, etc) when discussing significant relationships. Several relationships were glossed over, especially in section 3.3, which should be unpacked a bit more.

There is a potential role of genetic factors in both mitochondrial function and Alzheimer's Disease, including APOE. This should be discussed and assessed if measured. If it was not assessed, it should be added as a limitation.

Include justification for statistical adjustment for intracranial volume at age 70 for brain atrophy measures.

Reviewer #4

(Remarks to the Author)

Version 1:

Reviewer comments:

Reviewer #1

(Remarks to the Author)

We appreciate the Authors' time and effort in reviewing the manuscript. We feel that most of the comments have been appropriately addressed, improving the quality of the manuscript. Although we acknowledge the potential lack of power to address sex differences, including these results as exploratory is appreciated.

A few minor comments remain:

1. Thank you for adding more information to describe the samples in Table 1. The sub-sample with longitudinal data seemed significantly older than the cross-sectional cohort, and also less fit and more physically inactive (which could be age-related). However, there is no comment addressing these differences and the potential impact on the results/generalizability if the same sample were to be included in longitudinal analyses. Could some of the discrepancies in the number of significant results for the longitudinal analysis, but not cross-sectional, result simply from studying an older sample in which brain changes occur more rapidly? Also, there is no need to repeat information on the number of MRIs after the second time in the cross-sectional column of Table 1.

2. Please review the text on page 4, line 98, as the postcentral gyrus did not pass FDR correction.

Reviewer #2

(Remarks to the Author)

The authors adequately addressed my concerns.

Reviewer #3

(Remarks to the Author)

I feel that the critiques have been adequately addressed, no further comments.

Reviewer #4

(Remarks to the Author)

REVIEWER COMMENTS

Reviewer #1 (Remarks to the Author):

The submitted manuscript by Tian et al. examined the association between in vivo skeletal muscle mitochondrial function and multiple MRI measures cross-sectionally and longitudinally utilizing data from the Baltimore Longitudinal Study of Aging (BLSA). MRI/DTI outcomes included global and regional brain volumes, as well as white/gray matter microstructure (e.g., fractional anisotropy (FA), mean diffusivity (MD), radial diffusivity (RD), and axial diffusivity (AD)). The study found associations between higher skeletal muscle mitochondrial function and less atrophy over time in 13 structural ROIs after correction for multiple testing. The manuscript addresses an important area of research, as much work remains to be done to understand the role of mitochondrial function in brain aging. The paper is well-written, includes high-quality longitudinal imaging data from a diverse sample, and employs appropriate statistical methods. However, certain areas could be improved to enhance the clarity and impact of the work.

Thank you for your feedback. Please see our point-by-point responses to your comments below.

Major comments:

1. The authors state that skeletal muscle mitochondria oxidative capacity is a proxy for overall mitochondrial function, including in the brain. However, no information is provided to support this statement in the introduction. Skeletal muscle mitochondria oxidative capacity can simply reflect fitness, which is related to better neurological outcomes, and adjustment for self-reported physical activity may only account for a fraction of fitness. Can the authors provide additional information to support this statement? For example, are there studies correlating quadriceps and brain 31P-MRS data, or other experimental models? Expanding these points in the discussion would be appropriate to contextualize the results.

Responses: [Whether skeletal muscle mitochondrial oxidative capacity and mitochondrial function in the brain are directly related is currently unknown, and we present this idea as a hypothesis that needs to be tested in the future. As we point out above, skeletal muscle oxidative capacity is indeed associated with fitness \(Choi et al., 2016\). It is possible that higher skeletal muscle oxidative capacity reflects higher fitness level which is associated with better brain outcomes. To address the reviewer's comment, we performed sensitivity analyses by adjusting for the 400m walk time from the Long-Distance Corridor Walk or VO₂max from a maximal exercise treadmill test, which were both considered reliable tests of fitness. Of the 649 participants in the main analysis, 627 had data on 400m walk time and 586 had data on VO₂max. Longitudinal associations with brain volume changes remained substantially unchanged after adjusting for 400m walk time or VO₂max. Longitudinal associations with changes in microstructural integrity measures by DTI also remained similar after adjusting for 400m walk time but were attenuated after adjustment for VO₂ max. This attenuation may be due to the collinearity between kPCr and VO₂max and reduced sample size. We now provide these additional results as supplementary material. Please see Supplementary Table 1 and 3. Please also see the edited text: Statistical analysis: \"Because skeletal muscle oxidative capacity is associated with fitness \(Choi et al., 2016; Tian et al., 2022; Zane et al., 2017\), we performed sensitivity analyses by adjusting the longitudinal analysis for 400m walk time from the Long-Distance Corridor Walk or VO₂ max from a maximal exercise treadmill test, which were both considered reliable tests of fitness. Of the 649 participants in the main analysis, 627 had data on 400m walk and 586 had data on VO₂ max. Because physical activity and fitness were highly correlated, models adjusting for fitness did not include physical activity."](#) Results: "Longitudinal associations with brain volume change remained similar after adjusting for 400m walk time, VO₂max (Supplementary Table 1)... Longitudinal associations with changes in FA and MD of white matter remained similar after adjusting for 400m walk time ... but were attenuated after adjusting for VO₂max (Supplementary Table 3)." Discussion "While skeletal muscle oxidative capacity is related to fitness, the longitudinal associations between skeletal muscle oxidative capacity and brain atrophy were independent of concurrent fitness levels. Longitudinal associations with microstructural change persisted after accounting for the fitness measure

of 400m walk time but were attenuated after adjusting for VO₂ max. This attenuation is not surprising as fitness and vascular factors are strongly associated with white matter microstructure captured by DTI (Faulkner et al., 2024; Sexton et al., 2016; Tian et al., 2015; Williams et al., 2019). It is also possible that the collinearity between k_{PCr} and VO₂ max contributed to the attenuation of these associations. Because of the observational nature of this study, the detected longitudinal associations may shed light on but do not prove a causal relationship. In addition, we cannot exclude that better skeletal muscle oxidative capacity reflects in part the lifetime history of exercise and physical activity which may affect several aspects of brain health but may not be fully captured by the assessment of current fitness levels.”

To the best of our knowledge, no studies have investigated the relationship between skeletal muscle mitochondrial function and MRS data in the brain. We now add the discussion points. Please see the edited text: “Future studies are warranted to directly assess *in vivo* mitochondrial function in the brain via MR spectroscopy and examine its relationship with skeletal muscle oxidative capacity, which can further elucidate underlying mechanisms on mitochondria in the aging CNS and neurodegeneration.”

2. It would be helpful if the authors could provide a flowchart indicating how they arrived at the final sample of 649 participants. No information is provided on the number of participants who were excluded. Additionally, a table of participants’ characteristics presenting the original, cross-sectional, and longitudinal samples would help identify selection bias.

Responses: Following the reviewer’s suggestion, we now provide a flow chart of participant selection. Please see the new Figure 1. We also add information on participants with cross-sectional data only and participants with longitudinal data to Table 1. Please see the revised Table 1.

3. Were there any differences in the results by sex?

Responses: We did not focus on sex difference which could be a separate research topic. To answer the reviewer’s question, we tested sex differences in the associations between skeletal muscle oxidative capacity and brain structural change by adding an interaction term with sex. For most neuroimaging measures of MRI and DTI, there were no significant interaction terms with sex in longitudinal associations, except the volume of inferior temporal gyrus and MD of the corpus callosum and superior corona radiata, but none of these passed FDR-adjusted $p < 0.05$. For these measures with significant sex interactions at $p < 0.05$, we stratified the analyses by sex. The longitudinal association with inferior temporal gyrus volume and MD of the corpus callosum and superior corona radiata were significant or showed a trend in men ($p = 0.022$, $p = 0.052$, and $p = 0.099$, respectively), while they were not significant in women ($p = 0.406$, $p = 0.952$, and $p = 0.301$, respectively). We’d like to note that we feel that we are not powered to fully address sex differences due to the modest sample size, and these results should be considered with caution. We now add these findings to the manuscript as an exploratory analysis. Statistical analysis: “We explored sex differences by adding an interaction term with sex to the LME models.” Results: “In the exploratory analysis of sex differences, for most neuroimaging measures of MRI and DTI, there was no significant interaction term with sex in longitudinal associations, except MD of the cingulate part of the cingulum and superior corona radiata neither of which passed FDR-adjusted $p < 0.05$. For these 2 MD measures, we further stratified by sex. The longitudinal associations with MD of the cingulate part of the cingulum and superior corona radiata showed a trend in men ($p = 0.052$ and 0.099 , respectively), but were not significant in women ($p = 0.952$ and 0.301 , respectively).”

4. It would be interesting to include global DTI markers of FA and MD in total white matter — there may be additional signal than looking at individual tracts.

Responses: Following the reviewer’s comment, we now add results on global FA and MD in total white matter. Please see the edited text: “Cross-sectionally, higher k_{PCr} was associated with higher FA and lower MD in the total white matter ($p = 0.029$ and $p = 0.023$, respectively).” “Longitudinally, k_{PCr} was not associated with the change in the overall FA or MD in total white matter ($p = 0.064$ and $p = 0.141$, respectively).”

5. It is somewhat unusual to use DTI metrics in gray matter. FA and MD may not be as relevant in gray matter, given the more restricted and heterogeneous diffusion of water in cell bodies, as opposed to white matter axons. Can the authors provide a rationale for including gray matter FA and MD as outcomes of interest?

Responses: Following the reviewer's comment, we now provide a rationale for using FA and MD for gray matter regions. While the interpretation of MD is clearer for gray matter (i.e., mean diffusivity across various tissue components) and FA is clearer for white matter (i.e., anisotropy due to directional orientations of fibers), it is common to report significant results for both MD and FA. Significant findings can guide future research questions despite lower sensitivity in gray matter and challenges in interpretation. To clarify our rationale, we added the following text and reference: “Metrics of DTI, such as FA and MD, are widely used to quantify water molecules in white matter fiber tracks and can also provide quantitative measures of gray matter microstructure because of unconstrained water molecules in CSF and adjacent areas such as cerebral cortex (Abe et al., 2008).”

6. I recommend toning down conclusions on findings related to DTI outcomes, as these did not pass correction for multiple testing. Currently, the discussion suggests these are “robust”, and “strongly related” to mitochondrial function.

Responses: Following the reviewer's suggestion, we now revise the conclusions and discussion. We now remove any mention of “robust” and “strongly related” regarding DTI findings.

Minor comments:

7. The brain MRI methods section includes a paragraph in quotation, presumably because the same text has been used before. I would recommend either citing the source at the end of the paragraph.

Responses: Following the reviewer's suggestion, we now remove some of the cited text and cite the reference at the end of the paragraph.

8. Figure 3 is mentioned before Figure 2 in the text, so it needs to be reorganized in the manuscript.

Responses: Thank you for your comment. We now mention the Figure 3 following Figure 2. Because a flow chart figure is added, we now reorganized the number of figures: “Brain areas that showed significant longitudinal associations with \$k_{PCr}\$ and rates of change in top significant MRI and DTI markers by levels of \$k_{PCr}\$ are shown in Figure 4.”

9. Figure 1 does not show global measurements (SPARE-BA, SPARE-AD).

Responses: We now add results of global measures of SPARE-BA and SPARE-AD to Figure 1 (now Figure 2).

10. Figures 1 and 2 should clearly indicate that bold red data points correspond to FDR-corrected results in the legend.

Responses: Following the reviewer's comment, we now add a note to the legend of Figure 2 that “beyond red dotted lines are at \$p < 0.01\$ and also survived FDR- \$p < 0.05\$.” We note that none of the DTI associations were significant at FDR-adjusted $p < 0.05$.

11. Tables 2 and 3 could be added to the supplementary material as the same information is presented in Figures 1 and 2.

Responses: Tables 2 and 3 include statistics of beta and standard deviation, while Figures 1 and 2 are based on t-values. For this reason, we believe that it is important to retain Tables 2 and 3 in the main manuscript. Of course, we can remove them if needed.

12. Why do global and ventricular outcomes not have FDR-adjusted p-values in Table 2?

Responses: Following the reviewer's question, we performed FDR adjustment for all measures including global and ventricular measures. Longitudinal associations that survived FDR- $p < 0.05$ remained the same. Longitudinal associations with SPARE-BA and 4th ventricle also survived FDR- $p < 0.05$. We now revise accordingly. Please see the edited text: “The longitudinal associations with 4th ventricular volume and SPARE-BA scores also survived corrections for multiple testing using FDR (FDR- \$p < 0.05\$ ).”

13. Supplementary Table 1 has repeated headings; radial diffusivity is missing.

Responses: We now correct the heading in Supplementary Table 1 (now Supplementary Table 2).

Co-reviewed by Claudia Satizabal & Janette Vazquez

Reviewer #2 (Remarks to the Author):

The authors used data from the Baltimore Longitudinal Study of Aging (BLSA) to examine the association between in vivo skeletal muscle maximal oxidative capacity after exercise and longitudinal changes in brain structure assessed using imaging. While the study results represent an important contribution, the authors could contextualize better the mitochondria biology measure they are using and

how it relates to brain function (see specific comments below).

Introduction

- Line 51. “Mitochondrial function assessed in the skeletal muscle and blood predicts future cognitive impairment or dementia and is associated with cognitive function and biomarkers of AD and neurodegeneration (Tian, Bilgel, et al., 2023; Tian et al., 2024; Tian, Zweibaum, et al., 2023; Yang et al., 2021; Zhang et al., 2023).” Mitochondria have several functions, could you be specific about the mitochondrial outcomes measured in these studies?

Responses: Following the reviewer’s comment, we now add specific measures to this sentence. Please see the edited text: “Our recent findings show that *in vivo* skeletal muscle oxidative capacity via MR spectroscopy predicts the development of cognitive impairment or dementia and is associated with PET and blood biomarkers of AD and neurodegeneration (Tian, Bilgel, et al., 2023). Mitochondrial DNA copy number from blood samples predict future dementia and is associated with cognitive function (Tian, Zweibaum, et al., 2023; Yang et al., 2021; Zhang et al., 2023).”

- Line 55, what was the mitochondrial function assessed?

Responses: In this sentence, we refer to *in vivo* assessment of mitochondrial function. We now add clarity to this sentence. Please see the edited text: “However, existing data on how *in vivo* assessment of mitochondrial function is associated with brain structure are sparse.”

- Line 60 Do you have references to support that mitochondrial biology in skeletal muscle reflects brain mitochondria biology? Did any study look at the correlation between mitochondrial measures in muscle vs brain?

Responses: Thank you for your comment. Whether skeletal muscle mitochondrial oxidative capacity can be considered a proxy measure of brain mitochondrial function is unknown. Our current study demonstrates a longitudinal relationship between skeletal muscle oxidative capacity and brain structural changes. We propose potential mechanisms that can explain this association, including the possibility that skeletal muscle oxidative capacity reflects mitochondrial function in the brain since both the muscle and the brain are highly energy-demanding tissues. However, it is also possible (and suggested by a few studies) that myokines produced by active exercising muscle directly signal to the brain. Testing these hypotheses is an important area of research that we plan to address in the future. We have commented on potential mechanisms in the discussion and now add a reference: “Our findings highlight the importance of mitochondrial health in brain aging but also underline that the search for a mechanism explaining this connection warrants further investigation. One potential mechanism is that muscle mitochondrial function indicates general mitochondrial health and that muscle mitochondria can be considered a proxy measure of mitochondrial health across multiple tissues, including the brain. Another possibility is that the measure of oxidative capacity captures general muscle health and that positive signaling through soluble molecules and/or microvesicles may act in neurotrophic signaling that promotes brain health (Febbraio & Pedersen, 2020)”.

- Line 64, which function specifically? Could you introduce the measure you are using (skeletal muscle mitochondrial oxidative capacity) and explain which function it reflects? Why more of it after exercise is better?

Responses: Following the reviewer’s comment, we now add an introduction of the measure: “The exercise test used in this study is based on the theory of the creatine phosphate shuttle, a system that facilitates the transport of high energy phosphate from muscle cell mitochondria to myofibrils. At rest, ATP levels are high leading to the conversion of creatine to phosphocreatine. During exercise in the magnet, phosphocreatine breaks down to creatine and inorganic phosphate, which are monitored by MR. At the end of the exercise, mitochondria provide ATP to rephosphorylate creatine into creatine phosphate. The rate of rephosphorylation mostly depends on mitochondrial oxidative capacity and therefore this rate monitored by MRI is a good proxy measure of mitochondrial oxidative capacity (Conley et al., 2000).”

Method

- Has that measure of maximal oxidative capacity been used by other study? Does it correlate with other mitochondrial markers?

Responses: Yes, the skeletal muscle oxidative capacity assessed by P-31 MRS has been used and reported

in other studies (Amara et al., 2008; Conley et al., 2000; Cummings et al., 2023; Edwards et al., 2012). In the BLSA, we have found that skeletal muscle oxidative capacity is associated with mitochondrial DNA copy number from blood (Tian, Moore, et al., 2021). We now add this information to the methods. Please see the edited text: “This exercise protocol has been used in other studies (Amara et al., 2008; Cummings et al., 2023; Edwards et al., 2012). Skeletal muscle oxidative capacity is associated with mitochondrial DNA copy number from blood samples (Tian, Moore, et al., 2021).”

Results

- The individual datapoints are not shown, some illustrative scatterplots could be used for transparency.

Responses: Following the reviewer’s suggestion, we now provide individual datapoints. Please see supplementary Figure 2. “Estimated trajectories of top significant MRI and DTI markers at higher and lower levels of \$k_{PCr}\$ as well as individual datapoints are shown in Supplementary Figure 2.” We note that estimated trajectories are updated in the revised version to include covariate adjustment. Please see revised Figure 4.

Discussion

- 274-277: is there any prior evidence suggesting that skeletal muscle bioenergetics measures correlate with brain bioenergetics measures?

Responses: Please see our responses to a similar comment above. “Whether skeletal muscle mitochondrial oxidative capacity can be considered a proxy measure of brain mitochondrial function is unknown. Our current study demonstrates a longitudinal relationship between skeletal muscle oxidative capacity and brain structural changes. We propose potential mechanisms that can explain this association, including the possibility that skeletal muscle oxidative capacity reflects mitochondrial function in the brain since both the muscle and the brain are highly energy-demanding tissues. However, it is also possible (and suggested by a few studies) that myokines produced by active exercising muscle directly signal to the brain. Testing these hypotheses is an important area of research that we plan to address in the future.”

- Is it plausible that healthier individuals have better skeletal muscle bioenergetics and better brain health without the two being causality related?

Responses: The only way to establish causality is through intervention studies. Our longitudinal findings demonstrate that individuals with higher skeletal muscle oxidative capacity show preserved brain structure over a decade. Noteworthy, the cross-sectional associations are largely not significant. If the association was simply with better health, we would expect that individuals with higher skeletal muscle bioenergetics would have concurrent better brain health.

Reviewer #3 (Remarks to the Author):

This study leverages a large and well characterized cohort, the Baltimore Longitudinal Study of Aging, to examine the interesting and important relationship between skeletal muscle mitochondrial function and brain imaging outcomes. There are key strengths of the study including the inclusion of multiple imaging sequences and functional mitochondrial assessments. This type of study is needed and overdue, is novel, and contributes to the literature. However, there are several clarifications and additions that are needed to strengthen the manuscript.

There was no clear formal hypothesis presented in the introduction. What did the investigators expect to find?

Responses: The rationale that eventually led to performing this analysis was that the brain and the muscle (skeletal and cardiac) are the two most energetically demanding tissues in the human body. In addition, there was a sizeable literature suggesting that physical activity and exercise, the most important factor positively affecting mitochondrial mass and oxidative capacity, correlates with better brain health. Thus, we hypothesized that mitochondrial oxidative capacity would predict future brain health operationalized as brain atrophy and microstructural decline. At this stage, we did not hypothesize a specific mechanism and in the manuscript, we speculate that because muscle mitochondrial oxidative capacity is related to brain mitochondria oxidative capacity, there is some signaling from the muscle that may affect brain health. We now add information on related references: “We previously investigated cross-sectional associations between skeletal muscle oxidative capacity with brain PET imaging markers of amyloid

(Tian, Bilgel, et al., 2023) and longitudinal brain atrophy of dual decline in memory and gait related to mitochondrial dysfunction (Tian, Shardell, et al., 2023; Tian, Studenski, et al., 2021). The associations were found in several brain regions but strong in specific frontal and parietal areas, cingulate cortex, and cerebellum (Tian, Bilgel, et al., 2023; Tian, Studenski, et al., 2021).”

For cross-sectional measures, what was the mean length (and range) of durations between when the MRI scan was obtained and when the MR muscle functional mitochondrial assessment was performed? Based on the study design, there are differences in the length of follow up MRI scans based on age bins. Please clarify within the manuscript how many participants fell into each bin (younger than 60, 60-79, 80+)? Given that the first assessment of muscle mitochondrial function was used, there would have been varying frequency of MRI scans for each person during the longitudinal follow up period. How do the investigators feel this may affect the results (especially cross sectional findings)?

Responses: To clarify, for cross-sectional associations, measures of skeletal muscle oxidative capacity and brain MRI scans are obtained at the same BLSA visit, and data are collected within minutes in the MRI scanner. We now report the frequency of different age groups in the results. To address the reviewer’s question, we performed sensitivity analyses by limiting to participants aged 60 years or older, both MRI and DTI results of cross-sectional and longitudinal associations remained largely unchanged. Please see added text: “70% were aged 60 years or older.” Please also see added text: “We performed additional sensitivity analyses by limiting the analysis to participants aged 60 years or older” “Results remained largely unchanged when the analysis was limited to participants aged 60 years or older.”

For brain volume measures, did investigators assess hemisphere-specific differences (L vs R)? In addition to differences in cross sectional volumes between hemispheres in some brain regions, the rate of longitudinal atrophy can differ between hemispheres. Was handedness assessed?

Responses: We did not examine hemispheric differences in this study due to the already large number of measures we examined and the high intercorrelations among homologous hemispheric values. Doubling the number of outcomes would have resulted in reduced statistical power. With larger samples as the study continues, this can be further studied. Handedness was measured and we did not exclude any participants due to handedness. We now report handedness in the results: “91% were right-handed.” Given that multiple MRI scanners were used, details should be added for structural QC procedures, phantom scans, etc. that were run to ensure reproducibility across scanners. The only QC that was mentioned was related to the DTI.

Responses: Following the reviewer’s comment, we now note that the majority of scans were performed on a single 3T Philips Achieva scanner at the NIA MRI Facility. Before moving studies to this scanner in May 2009, approximately 20 participants were studied on both the KKI and NIA scanners, and there was low variability and no consistent differences between scanners with respect to MRI volumes.

Additionally, a detailed and validated harmonization approach has been incorporated into the analysis of structural MRI scans (Erus et al., 2018). We have added information on the longitudinal reliability of volumetric and DTI measures. Please see the edited text: “Longitudinal measures of regional brain volumes by MRI have good stability and consistency with a previously reported intraclass correlation between 0.89 and 0.99 (Armstrong et al., 2019).” “Longitudinal measures of brain DTI measures of FA and MD show average intraclass correlations of 0.76 and 0.66, respectively, and within scanner versus between scanner intraclass correlations were comparable for the measures at 3T (Venkatraman et al., 2015).”

Results section: please include statistics (i.e. p values, regression coefficients, etc) when discussing significant relationships. Several relationships were glossed over, especially in section 3.3, which should be unpacked a bit more.

Responses: Following the reviewer’s comment, we now add statistical details to the results section, including p-values, FDR-adjusted p-values, and regression coefficients, especially section 3.3.

There is a potential role of genetic factors in both mitochondrial function and Alzheimer’s Disease, including APOE. This should be discussed and assessed if measured. If it was not assessed, it should be added as a limitation.

Responses: Thank you for your comment. We performed additional analyses by adjusting for APOE e4

carrier status and the results did not change. We now add results and discussion points to the manuscript and add Supplementary Table x. Please see the edited text: “We performed additional sensitivity analyses by adjusting for apolipoprotein E ε4 (APOE ε4) carrier status.” “Results remained similar after adjusting for APOE ε4 carrier status (Supplementary Table 1) (Supplementary Table 3)”. We also checked the relationship between skeletal muscle oxidative capacity and APOE e4 status. kPCr was not related to APOE e4 carrier status ($t=1.113$, $p=0.267$).

Include justification for statistical adjustment for intracranial volume at age 70 for brain atrophy measures.

Responses: Following the reviewer’s comment, we now add the justification for using intracranial volume at age 70. Please see the edited text: “Adjustment for a single baseline measure of ICV was to avoid any effect of potential changes on ICV measures over time (Caspi et al., 2020). Age 70 was around the average age of the BLSA population at the baseline MRI.”

Reviewer #4 (Remarks to the Author):

References added to the revision:

- Abe, O., Yamasue, H., Aoki, S., Suga, M., Yamada, H., Kasai, K., Masutani, Y., Kato, N., Kato, N., & Ohtomo, K. (2008). Aging in the CNS: comparison of gray/white matter volume and diffusion tensor data. *Neurobiol Aging*, 29(1), 102-116. <https://doi.org/10.1016/j.neurobiolaging.2006.09.003>
- Amara, C. E., Marcinek, D. J., Shankland, E. G., Schenkman, K. A., Arakaki, L. S., & Conley, K. E. (2008). Mitochondrial function in vivo: spectroscopy provides window on cellular energetics. *Methods*, 46(4), 312-318. <https://doi.org/10.1016/j.ymeth.2008.10.001>
- Armstrong, N. M., An, Y., Beason-Held, L., Doshi, J., Erus, G., Ferrucci, L., Davatzikos, C., & Resnick, S. M. (2019). Predictors of neurodegeneration differ between cognitively normal and subsequently impaired older adults. *Neurobiol Aging*, 75, 178-186. <https://doi.org/10.1016/j.neurobiolaging.2018.10.024>
- Caspi, Y., Brouwer, R. M., Schnack, H. G., van de Nieuwenhuijzen, M. E., Cahn, W., Kahn, R. S., Niessen, W. J., van der Lugt, A., & Pol, H. H. (2020). Changes in the intracranial volume from early adulthood to the sixth decade of life: A longitudinal study. *Neuroimage*, 220, 116842. <https://doi.org/10.1016/j.neuroimage.2020.116842>
- Choi, S., Reiter, D. A., Shardell, M., Simonsick, E. M., Studenski, S., Spencer, R. G., Fishbein, K. W., & Ferrucci, L. (2016). 31P Magnetic Resonance Spectroscopy Assessment of Muscle Bioenergetics as a Predictor of Gait Speed in the Baltimore Longitudinal Study of Aging. *J Gerontol A Biol Sci Med Sci*, 71(12), 1638-1645. <https://doi.org/10.1093/gerona/glw059>
- Conley, K. E., Jubrias, S. A., & Esselman, P. C. (2000). Oxidative capacity and ageing in human muscle. *J Physiol*, 526 Pt 1(Pt 1), 203-210. <https://doi.org/10.1111/j.1469-7793.2000.t01-1-00203.x>
- Cummings, S. R., Newman, A. B., Coen, P. M., Hepple, R. T., Collins, R., Kennedy Ms, K., Danielson, M., Peters, K., Blackwell, T., Johnson, E., Mau, T., Shankland, E. G., Lui, L. Y., Patel, S., Young, D., Glynn, N. W., Strotmeyer, E. S., Esser, K. A., Marcinek, D. J., Goodpaster, B. H., Kritchevsky, S., & Cawthon, P. M. (2023). The Study of Muscle, Mobility and Aging (SOMMA): A Unique Cohort Study About the Cellular Biology of Aging and Age-related Loss of Mobility. *J Gerontol A Biol Sci Med Sci*, 78(11), 2083-2093. <https://doi.org/10.1093/gerona/glad052>
- Edwards, L. M., Tyler, D. J., Kemp, G. J., Dwyer, R. M., Johnson, A., Holloway, C. J., Nevill, A. M., & Clarke, K. (2012). The reproducibility of 31-phosphorus MRS measures of muscle energetics at 3 Tesla in trained men. *PLoS One*, 7(6), e37237. <https://doi.org/10.1371/journal.pone.0037237>
- Erus, G., Doshi, J., An, Y., Verganelakis, D., Resnick, S. M., & Davatzikos, C. (2018). Longitudinally and inter-site consistent multi-atlas based parcellation of brain anatomy using harmonized atlases. *Neuroimage*, 166, 71-78. <https://doi.org/10.1016/j.neuroimage.2017.10.026>
- Faulkner, M. E., Gong, Z., Bilgel, M., Laporte, J. P., Guo, A., Bae, J., Palchamy, E., Kaileh, M., Bergeron, C. M., Bergeron, J., Church, S., D'Agostino, J., Ferrucci, L., & Bouhrara, M. (2024). Evidence of association between higher cardiorespiratory fitness and higher cerebral myelination in aging. *Proc Natl Acad Sci U S A*, 121(35), e2402813121. <https://doi.org/10.1073/pnas.2402813121>
- Febbraio, M. A., & Pedersen, B. K. (2020). Who would have thought - myokines two decades on. *Nat Rev Endocrinol*, 16(11), 619-620. <https://doi.org/10.1038/s41574-020-00408-7>
- Sexton, C. E., Betts, J. F., Demnitz, N., Dawes, H., Ebmeier, K. P., & Johansen-Berg, H. (2016). A systematic review of MRI studies examining the relationship between physical fitness and activity and the white matter of the ageing brain. *Neuroimage*, 131, 81-90. <https://doi.org/10.1016/j.neuroimage.2015.09.071>
- Tian, Q., Bilgel, M., Walker, K. A., Moghekar, A. R., Fishbein, K. W., Spencer, R. G., Resnick, S. M., & Ferrucci, L. (2023). Skeletal muscle mitochondrial function predicts cognitive impairment and is associated with biomarkers of Alzheimer's disease and neurodegeneration. *Alzheimers Dement*, 19(10), 4436-4445. <https://doi.org/10.1002/alz.13388>
- Tian, Q., Glynn, N. W., Erickson, K. I., Aizenstein, H. J., Simonsick, E. M., Yaffe, K., Harris, T. B., Kritchevsky, S. B., Boudreau, R. M., Newman, A. B., Lopez, O. L., Saxton, J., & Rosano, C. (2015). Objective measures of physical activity, white matter integrity and cognitive status in adults over age 80. *Behav Brain Res*, 284, 51-57. <https://doi.org/10.1016/j.bbr.2015.01.045>

Tian, Q., Mitchell, B. A., Zampino, M., Fishbein, K. W., Spencer, R. G., & Ferrucci, L. (2022). Muscle mitochondrial energetics predicts mobility decline in well-functioning older adults: The Baltimore longitudinal study of aging. *Aging Cell*, 21(2), e13552. <https://doi.org/10.1111/accel.13552>

Tian, Q., Moore, A. Z., Oppong, R., Ding, J., Zampino, M., Fishbein, K. W., Spencer, R. G., & Ferrucci, L. (2021). Mitochondrial DNA copy number and heteroplasmy load correlate with skeletal muscle oxidative capacity by P31 MR spectroscopy. *Aging Cell*, 20(11), e13487. <https://doi.org/10.1111/accel.13487>

Tian, Q., Shardell, M. D., Kuo, P. L., Tanaka, T., Simonsick, E. M., Moaddel, R., Resnick, S. M., & Ferrucci, L. (2023). Plasma metabolomic signatures of dual decline in memory and gait in older adults. *Geroscience*, 45(4), 2659-2667. <https://doi.org/10.1007/s11357-023-00792-8>

Tian, Q., Studenski, S. A., Montero-Odasso, M., Davatzikos, C., Resnick, S. M., & Ferrucci, L. (2021). Cognitive and neuroimaging profiles of older adults with dual decline in memory and gait speed. *Neurobiol Aging*, 97, 49-55. <https://doi.org/10.1016/j.neurobiolaging.2020.10.002>

Tian, Q., Zweibaum, D., Qian, Y., Pilling, L. C., Casanova, F., Atkins, J. L., Melzer, D., Ding, J., & Ferrucci, L. (2023). Mitochondrial DNA copy number predicts dementia risk with early mobility impairment. *Alzheimer's & Dementia*, 19(S12). <https://doi.org/10.1002/alz.077229>

Venkatraman, V. K., Gonzalez, C. E., Landman, B., Goh, J., Reiter, D. A., An, Y., & Resnick, S. M. (2015). Region of interest correction factors improve reliability of diffusion imaging measures within and across scanners and field strengths. *Neuroimage*, 119, 406-416. <https://doi.org/10.1016/j.neuroimage.2015.06.078>

Williams, O. A., An, Y., Beason-Held, L., Huo, Y., Ferrucci, L., Landman, B. A., & Resnick, S. M. (2019). Vascular burden and APOE epsilon4 are associated with white matter microstructural decline in cognitively normal older adults. *Neuroimage*, 188, 572-583. <https://doi.org/10.1016/j.neuroimage.2018.12.009>

Yang, S. Y., Castellani, C. A., Longchamps, R. J., Pillalamarri, V. K., O'Rourke, B., Guallar, E., & Arking, D. E. (2021). Blood-derived mitochondrial DNA copy number is associated with gene expression across multiple tissues and is predictive for incident neurodegenerative disease. *Genome Res*, 31(3), 349-358. <https://doi.org/10.1101/gr.269381.120>

Zane, A. C., Reiter, D. A., Shardell, M., Cameron, D., Simonsick, E. M., Fishbein, K. W., Studenski, S. A., Spencer, R. G., & Ferrucci, L. (2017). Muscle strength mediates the relationship between mitochondrial energetics and walking performance. *Aging Cell*, 16(3), 461-468. <https://doi.org/10.1111/accel.12568>

Zhang, Y., Liu, X., Wiggins, K. L., Kurniansyah, N., Guo, X., Rodrigue, A. L., Zhao, W., Yanek, L. R., Ratliff, S. M., Pitsillides, A., Aguirre Patino, J. S., Sofer, T., Arking, D. E., Austin, T. R., Beiser, A. S., Blangero, J., Boerwinkle, E., Bressler, J., Curran, J. E., Hou, L., Hughes, T. M., Kardia, S. L. R., Launer, L. J., Levy, D., Mosley, T. H., Nasrallah, I. M., Rich, S. S., Rotter, J. I., Seshadri, S., Tarraf, W., Gonzalez, K. A., Ramachandran, V., Yaffe, K., Nyquist, P. A., Psaty, B. M., DeCarli, C. S., Smith, J. A., Glahn, D. C., Gonzalez, H. M., Bis, J. C., Fornage, M., Heckbert, S. R., Fitzpatrick, A. L., Liu, C., Satizabal, C. L., NHLBI Trans-Omics for Precision Medicine program, M., & Neurocognitive Working, G. (2023). Association of Mitochondrial DNA Copy Number With Brain MRI Markers and Cognitive Function: A Meta-analysis of Community-Based Cohorts. *Neurology*, 100(18), e1930-e1943. <https://doi.org/10.1212/WNL.0000000000207157>

REVIEWERS' COMMENTS

Reviewer #1 (Remarks to the Author):

We appreciate the Authors' time and effort in reviewing the manuscript. We feel that most of the comments have been appropriately addressed, improving the quality of the manuscript.

Although we acknowledge the potential lack of power to address sex differences, including these results as exploratory is appreciated.

A few minor comments remain:

1. Thank you for adding more information to describe the samples in Table 1. The sub-sample with longitudinal data seemed significantly older than the cross-sectional cohort, and also less fit and more physically inactive (which could be age-related). However, there is no comment addressing these differences and the potential impact on the results/generalizability if the same sample were to be included in longitudinal analyses. Could some of the discrepancies in the number of significant results for the longitudinal analysis, but not cross-sectional, result simply from studying an older sample in which brain changes occur more rapidly? Also, there is no need to repeat information on the number of MRIs after the second time in the cross-sectional column of Table 1.

Responses: Thank you for your comments. Because 70% of the study sample are aged 60 years or older, results are mainly from older participants. As the study is ongoing, we can further investigate this association in younger samples when more data become available. Please see added text in the discussion (2nd paragraph on page 5): "Because the majority of the study participants are aged 60 years or older, whether this relationship persists in younger age requires future research." In Table 1, we now remove the number of MRIs after the second time in the cross-sectional column.

2. Please review the text on page 4, line 98, as the postcentral gyrus did not pass FDR correction.

Responses: Thank you for pointing this out. We now correct this sentence by removing postcentral gyrus.

Reviewer #2 (Remarks to the Author):

The authors adequately addressed my concerns.

Thank you.

Reviewer #3 (Remarks to the Author):

I feel that the critiques have been adequately addressed, no further comments.

Thank you.

Reviewer #4 (Remarks to the Author):

Thank you.